# Rethinking LDA: Moment Matching for Discrete ICA

**Anastasia Podosinnikova**     **Francis Bach**     **Simon Lacoste-Julien**
INRIA - École normale supérieure Paris

## Abstract

We consider moment matching techniques for estimation in latent Dirichlet allocation (LDA). By drawing explicit links between LDA and discrete versions of independent component analysis (ICA), we first derive a new set of cumulant-based tensors, with an improved sample complexity. Moreover, we reuse standard ICA techniques such as joint diagonalization of tensors to improve over existing methods based on the tensor power method. In an extensive set of experiments on both synthetic and real datasets, we show that our new combination of tensors and orthogonal joint diagonalization techniques outperforms existing moment matching methods.

## 1   Introduction

Topic models have emerged as flexible and important tools for the modelisation of text corpora. While early work has focused on graphical-model approximate inference techniques such as variational inference [1] or Gibbs sampling [2], tensor-based moment matching techniques have recently emerged as strong competitors due to their computational speed and theoretical guarantees [3, 4]. In this paper, we draw explicit links with the independent component analysis (ICA) literature (e.g., [5] and references therein) by showing a strong relationship between latent Dirichlet allocation (LDA) [1] and ICA [6, 7, 8]. We can then reuse standard ICA techniques and results, and derive new tensors with better sample complexity and new algorithms based on joint diagonalization.

## 2   Is LDA discrete PCA or discrete ICA?

**Notation.** Following the text modeling terminology, we define a corpus $X = \{x_1, \ldots, x_N\}$ as a collection of $N$ documents. Each document is a collection $\{w_{n1}, \ldots, w_{nL_n}\}$ of $L_n$ tokens. It is convenient to represent the $\ell$-th token of the $n$-th document as a 1-of-$M$ encoding with an indicator vector $w_{n\ell} \in \{0, 1\}^M$ with only one non-zero, where $M$ is the vocabulary size, and each document as the count vector $x_n := \sum_\ell w_{n\ell} \in \mathbb{R}^M$. In such representation, the length $L_n$ of the $n$-th document is $L_n = \sum_m x_{nm}$. We will always use index $k \in \{1, \ldots, K\}$ to refer to topics, index $n \in \{1, \ldots, N\}$ to refer to documents, index $m \in \{1, \ldots, M\}$ to refer to words from the vocabulary, and index $\ell \in \{1, \ldots, L_n\}$ to refer to tokens of the $n$-th document. The plate diagrams of the models from this section are presented in Appendix A.

**Latent Dirichlet allocation** [1] is a generative probabilistic model for discrete data such as text corpora. In accordance to this model, the $n$-th document is modeled as an *admixture* over the vocabulary of $M$ words with $K$ latent topics. Specifically, the latent variable $\theta_n$, which is sampled from the Dirichlet distribution, represents the topic mixture proportion over $K$ topics for the $n$-th document. Given $\theta_n$, the topic choice $z_{n\ell}|\theta_n$ for the $\ell$-th token is sampled from the multinomial distribution with the probability vector $\theta_n$. The token $w_{n\ell}|z_{n\ell}, \theta_n$ is then sampled from the multinomial distribution with the probability vector $d_{z_{n\ell}}$, or $d_k$ if $k$ is the index of the non-zero element in $z_{n\ell}$. This vector $d_k$ is the $k$-th topic, that is a vector of probabilities over the words from the vocabulary subject to the simplex constraint, i.e., $d_k \in \Delta_M$, where $\Delta_M := \{d \in \mathbb{R}^M : d \succeq 0, \sum_m d_m = 1\}$. This generative process of a document (the index $n$ is omitted for simplicity) can be summarized as

$$\theta \sim \text{Dirichlet}(c),$$
$$z_\ell | \theta \sim \text{Multinomial}(1, \theta), \qquad \qquad (1)$$
$$w_\ell | z_\ell, \theta \sim \text{Multinomial}(1, d_{z_\ell}).$$

One can think of the latent variables $z_\ell$ as auxiliary variables which were introduced for convenience of inference, but can in fact be marginalized out [9], which leads to the following model

$$\theta \sim \text{Dirichlet}(c),$$
$$x | \theta \sim \text{Multinomial}(L, D\theta), \qquad \text{LDA model} \quad (2)$$

where $D \in \mathbb{R}^{M \times K}$ is the topic matrix with the $k$-th column equal to the $k$-th topic $d_k$, and $c \in \mathbb{R}_{++}^K$ is the vector of parameters for the Dirichlet distribution. While a document is represented as a set of tokens $w_\ell$ in the formulation (1), the formulation (2) instead compactly represents a document as the count vector $x$. Although the two representations are equivalent, we focus on the second one in this paper and therefore refer to it as the LDA model.

Importantly, the LDA model does not model the length of documents. Indeed, although the original paper [1] proposes to model the document length as $L|\lambda \sim \text{Poisson}(\lambda)$, this is never used in practice and, in particular, the parameter $\lambda$ is not learned. Therefore, in the way that the LDA model is typically used, it does not provide a complete generative process of a document as there is no rule to sample $L|\lambda$. In this paper, this fact is important, as we need to model the document length in order to make the link with discrete ICA.

**Discrete PCA.** The LDA model (2) can be seen as a discretization of principal component analysis (PCA) via replacement of the normal likelihood with the multinomial one and adjusting the prior [9] in the following probabilistic PCA model [10, 11]: $\theta \sim \text{Normal}(0, I_K)$ and $x|\theta \sim \text{Normal}(D\theta, \sigma^2 I_M)$, where $D \in \mathbb{R}^{M \times K}$ is a transformation matrix and $\sigma$ is a parameter.

**Discrete ICA (DICA).** Interestingly, a small extension of the LDA model allows its interpretation as a discrete independent component analysis model. The extension naturally arises when the document length for the LDA model is modeled as a random variable from the gamma-Poisson mixture (which is equivalent to a negative binomial random variable), i.e., $L|\lambda \sim \text{Poisson}(\lambda)$ and $\lambda \sim \text{Gamma}(c_0, b)$, where $c_0 := \sum_k c_k$ is the shape parameter and $b > 0$ is the rate parameter. The LDA model (2) with such document length is equivalent (see Appendix B.1) to

$$\alpha_k \sim \text{Gamma}(c_k, b),$$
$$x_m | \alpha \sim \text{Poisson}([D\alpha]_m), \qquad \text{GP model} \quad (3)$$

where all $\alpha_1, \alpha_2, \ldots, \alpha_K$ are mutually independent, the parameters $c_k$ coincide with the ones of the LDA model in (2), and the free parameter $b$ can be seen (see Appendix B.2) as a scaling parameter for the document length when $c_0$ is already prescribed.

This model was introduced by Canny [12] and later named as a discrete ICA model [13]. It is more natural, however, to name model (3) as the gamma-Poisson (GP) model and the model

$$\alpha_1, \ldots, \alpha_K \sim \text{mutually independent},$$
$$x_m | \alpha \sim \text{Poisson}([D\alpha]_m) \qquad \text{DICA model} \quad (4)$$

as the discrete ICA (DICA) model. The only difference between (4) and the standard ICA model [6, 7, 8] (without additive noise) is the presence of the Poisson noise which enforces discrete, instead of continuous, values of $x_m$. Note also that (a) the discrete ICA model is a *semi-parametric* model that can adapt to any distribution on the topic intensities $\alpha_k$ and that (b) the GP model (3) is a particular case of both the LDA model (2) and the DICA model (4).

Thanks to this close connection between LDA and ICA, we can reuse standard ICA techniques to derive new efficient algorithms for topic modeling.

## 3 Moment matching for topic modeling

The method of moments estimates latent parameters of a probabilistic model by matching theoretical expressions of its moments with their sample estimates. Recently [3, 4], the method of moments was applied to different latent variable models including LDA, resulting in computationally fast

learning algorithms with theoretical guarantees. For LDA, they (a) construct *LDA moments* with a particular diagonal structure and (b) develop algorithms for estimating the parameters of the model by exploiting this diagonal structure. In this paper, we introduce novel *GP/DICA cumulants* with a similar to the LDA moments structure. This structure allows to reapply the algorithms of [3, 4] for the estimation of the model parameters, with the same theoretical guarantees. We also consider another algorithm applicable to both the LDA moments and the GP/DICA cumulants.

## 3.1 Cumulants of the GP and DICA models

In this section, we derive and analyze the novel cumulants of the DICA model. As the GP model is a particular case of the DICA model, all results of this section extend to the GP model.

The first three *cumulant tensors* for the random vector $x$ can be defined as follows

$$\operatorname{cum}(x) := \mathbb{E}(x), \tag{5}$$

$$\operatorname{cum}(x, x) := \operatorname{cov}(x, x) = \mathbb{E}\left[(x - \mathbb{E}(x))(x - \mathbb{E}(x))^\top\right], \tag{6}$$

$$\operatorname{cum}(x, x, x) := \mathbb{E}\left[(x - \mathbb{E}(x)) \otimes (x - \mathbb{E}(x)) \otimes (x - \mathbb{E}(x))\right], \tag{7}$$

where $\otimes$ denotes the tensor product (see some properties of cumulants in Appendix C.1). The essential property of the cumulants (which does not hold for moments) that we use in this paper is that the cumulant tensor for a random vector with *independent* components is *diagonal*.

Let $y = D\alpha$; then for the Poisson random variable $x_m | y_m \sim \operatorname{Poisson}(y_m)$, the expectation is $\mathbb{E}(x_m | y_m) = y_m$. Hence, by the law of total expectation and the linearity of expectation, the expectation in (5) has the following form

$$\mathbb{E}(x) = \mathbb{E}(\mathbb{E}(x|y)) = \mathbb{E}(y) = D\mathbb{E}(\alpha). \tag{8}$$

Further, the variance of the Poisson random variable $x_m$ is $\operatorname{var}(x_m|y_m) = y_m$ and, as $x_1$, $x_2$, ..., $x_M$ are conditionally independent given $y$, then their covariance matrix is diagonal, i.e., $\operatorname{cov}(x, x|y) = \operatorname{diag}(y)$. Therefore, by the law of total covariance, the covariance in (6) has the form

$$\operatorname{cov}(x, x) = \mathbb{E}\left[\operatorname{cov}(x, x|y)\right] + \operatorname{cov}\left[\mathbb{E}(x|y), \mathbb{E}(x|y)\right]$$
$$= \operatorname{diag}\left[\mathbb{E}(y)\right] + \operatorname{cov}(y, y) = \operatorname{diag}\left[\mathbb{E}(x)\right] + D\operatorname{cov}(\alpha, \alpha)D^\top, \tag{9}$$

where the last equality follows by the multilinearity property of cumulants (see Appendix C.1). Moving the first term from the RHS of (9) to the LHS, we define

$$S := \operatorname{cov}(x, x) - \operatorname{diag}\left[\mathbb{E}(x)\right]. \qquad \text{DICA S-cum.} \tag{10}$$

From (9) and by the independence of $\alpha_1, \ldots, \alpha_K$ (see Appendix C.3), $S$ has the following diagonal structure

$$S = \sum_k \operatorname{var}(\alpha_k)d_k d_k^\top = D\operatorname{diag}\left[\operatorname{var}(\alpha)\right]D^\top. \tag{11}$$

By analogy with the second order case, using the law of total cumulance, the multilinearity property of cumulants, and the independence of $\alpha_1, \ldots, \alpha_K$, we derive in Appendix C.2 expression (24), similar to (9), for the third cumulant (7). Moving the terms in this expression, we define a tensor $T$ with the following element

$$[T]_{m_1 m_2 m_3} := \operatorname{cum}(x_{m_1}, x_{m_2}, x_{m_3}) + 2\delta(m_1, m_2, m_3)\mathbb{E}(x_{m_1}) \qquad \text{DICA T-cum.} \tag{12}$$
$$- \delta(m_2, m_3)\operatorname{cov}(x_{m_1}, x_{m_2}) - \delta(m_1, m_3)\operatorname{cov}(x_{m_1}, x_{m_2}) - \delta(m_1, m_2)\operatorname{cov}(x_{m_1}, x_{m_3}),$$

where $\delta$ is the Kronecker delta. By analogy with (11) (Appendix C.3), the diagonal structure of tensor $T$:

$$T = \sum_k \operatorname{cum}(\alpha_k, \alpha_k, \alpha_k)d_k \otimes d_k \otimes d_k. \tag{13}$$

In Appendix E.1, we recall (in our notation) the matrix $S$ (39) and the tensor $T$ (40) for the LDA model [3], which are analogues of the matrix $S$ (10) and the tensor $T$ (12) for the GP/DICA models. Slightly abusing terminology, we refer to the matrix $S$ (39) and the tensor $T$ (40) as the *LDA moments* and to the matrix $S$ (10) and the tensor $T$ (12) as the *GP/DICA cumulants*. The diagonal structure (41) & (42) of the LDA moments is similar to the diagonal structure (11) & (13) of the GP/DICA cumulants, though arising through a slightly different argument, as discussed at the end of

Appendix E.1. Importantly, due to this similarity, the algorithmic frameworks for both the GP/DICA cumulants and the LDA moments coincide.

The following sample complexity results apply to the sample estimates of the GP cumulants:[1]

**Proposition 3.1.** *Under the GP model, the expected error for the sample estimator $\widehat{S}$ (29) for the GP cumulant $S$ (10) is:*

$$\mathbb{E}\left[\|\widehat{S} - S\|_F\right] \leq \sqrt{\mathbb{E}\left[\|\widehat{S} - S\|_F^2\right]} \leq O\left(\frac{1}{\sqrt{N}} \max\left[\Delta \bar{L}^2, \bar{c}_0 \bar{L}\right]\right), \tag{14}$$

*where* $\Delta := \max_k \|d_k\|_2^2$, $\bar{c}_0 := \min(1, c_0)$ *and* $\bar{L} := \mathbb{E}(L)$.

A high probability bound could be derived using concentration inequalities for Poisson random variables [14]; but the expectation already gives the right order of magnitude for the error (for example via Markov's inequality). The expression (29) for an unbiased finite sample estimate $\widehat{S}$ of $S$ and the expression (30) for an unbiased finite sample estimate $\widehat{T}$ of $T$ are defined[2] in Appendix C.4. A sketch of a proof for Proposition 3.1 can be found in Appendix D.

By following a similar analysis as in [15], we can rephrase the topic recovery error in term of the error on the GP cumulant. Importantly, the whitening transformation (introduced in Section 4) redivides the error on $S$ (14) by $\bar{L}^2$, which is the scale of $S$ (see Appendix D.5 for details). This means that the contribution from $\hat{S}$ to the recovery error will scale as $O(1/\sqrt{N} \max\{\Delta, \bar{c}_0/\bar{L}\})$, where both $\Delta$ and $\bar{c}_0/\bar{L}$ are smaller than 1 and can be very small. We do not present the exact expression for the expected squared error for the estimator of $T$, but due to a similar structure in the derivation, we expect the analogous bound of $\mathbb{E}[\|\widehat{T} - T\|_F] \leq 1/\sqrt{N} \max\{\Delta^{3/2}\bar{L}^3, \bar{c}_0^{3/2}\bar{L}^{3/2}\}$.

Current sample complexity results of the LDA moments [3] can be summarized as $O(1/\sqrt{N})$. However, the proof (which can be found in the supplementary material [15]) analyzes only the case when finite sample estimates of the LDA moments are constructed from *one* triple per document, i.e., $w_1 \otimes w_2 \otimes w_3$ only, and not from the U-statistics that average multiple (dependent) triples per document as in the practical expressions (43) and (44). Moreover, one has to be careful when comparing upper bounds. Nevertheless, comparing the bound (14) with the current theoretical results for the LDA moments, we see that the GP/DICA cumulants sample complexity contains the $\ell_2$-norm of the columns of the topic matrix $D$ in the numerator, as opposed to the $O(1)$ coefficient for the LDA moments. This norm can be significantly smaller than 1 for vectors in the simplex (e.g., $\Delta = O(1/\|d_k\|_0)$ for sparse topics). This suggests that the GP/DICA cumulants may have better finite sample convergence properties than the LDA moments and our experimental results in Section 5.2 are indeed consistent with this statement.

The GP/DICA cumulants have a somewhat more intuitive derivation than the LDA moments as they are expressed via the count vectors $x$ (which are the sufficient statistics for the model) and not the tokens $w_\ell$'s. Note also that the construction of the LDA moments depend on the unknown parameter $c_0$. Given that we are in an unsupervised setting and that moreover the evaluation of LDA is a difficult task [16], setting this parameter is non-trivial. In Appendix G.4, we observe experimentally that the LDA moments are somewhat sensitive to the choice of $c_0$.

## 4 Diagonalization algorithms

How is the diagonal structure (11) of $S$ and (13) of $T$ going to be helpful for the estimation of the model parameters? This question has already been thoroughly investigated in the signal processing (see, e.g., [17, 18, 19, 20, 21, 5] and references therein) and machine learning (see [3, 4] and references therein) literature. We review the approach in this section. Due to similar diagonal structure, the algorithms of this section apply to both the LDA moments and the GP/DICA cumulants.

For simplicity, let us rewrite expressions (11) and (13) for $S$ and $T$ as follows

$$S = \sum_k s_k d_k d_k^\top, \qquad T = \sum_k t_k d_k \otimes d_k \otimes d_k, \tag{15}$$

where $s_k := \text{var}(\alpha_k)$ and $t_k := \text{cum}(\alpha_k, \alpha_k, \alpha_k)$. Introducing the rescaled topics $\widetilde{d}_k := \sqrt{s_k} d_k$, we can also rewrite $S = \widetilde{D}\widetilde{D}^\top$. Following the same assumption from [3] that the topic vectors are linearly independent ($\widetilde{D}$ is full rank), we can compute a whitening matrix $W \in \mathbb{R}^{K \times M}$ of $S$, i.e., a matrix such that $WSW^\top = I_K$ where $I_K$ is the $K$-by-$K$ identity matrix (see Appendix F.1 for more details). As a result, the vectors $z_k := W\widetilde{d}_k$ form an orthonormal set of vectors.

Further, let us define a projection $\mathcal{T}(v) \in \mathbb{R}^{K \times K}$ of a tensor $\mathcal{T} \in \mathbb{R}^{K \times K \times K}$ onto a vector $u \in \mathbb{R}^K$:

$$\mathcal{T}(u)_{k_1 k_2} := \sum\nolimits_{k_3} \mathcal{T}_{k_1 k_2 k_3} u_{k_3}. \tag{16}$$

Applying the multilinear transformation (see, e.g., [4] for the definition) with $W^\top$ to the tensor $T$ from (15) and projecting the resulting tensor $\mathcal{T} := T(W^\top, W^\top, W^\top)$ onto some vector $u \in \mathbb{R}^K$, we obtain

$$\mathcal{T}(u) = \sum\nolimits_k \widetilde{t}_k \langle z_k, u \rangle z_k z_k^\top, \tag{17}$$

where $\widetilde{t}_k := t_k / s_k^{3/2}$ is due to the rescaling of topics and $\langle \cdot, \cdot \rangle$ stands for the inner product. As the vectors $z_k$ are orthonormal, the pairs $z_k$ and $\lambda_k := \widetilde{t}_k \langle z_k, u \rangle$ are eigenpairs of the matrix $\mathcal{T}(u)$, which are uniquely defined if the eigenvalues $\lambda_k$ are all different. If they are unique, we can recover the GP/DICA (as well as LDA) model parameters via $\widetilde{d}_k = W^\dagger z_k$ and $\widetilde{t}_k = \lambda_k / \langle z_k, u \rangle$.

This procedure was referred to as the spectral algorithm for LDA [3] and the fourth-order[3] blind identification algorithm for ICA [17, 18]. Indeed, one can expect that the finite sample estimates $\widehat{S}$ (29) and $\widehat{T}$ (30) possess approximately the diagonal structure (11) and (13) and, therefore, the reasoning from above can be applied, assuming that the effect of the sampling error is controlled.

This spectral algorithm, however, is known to be quite unstable in practice (see, e.g., [22]). To overcome this problem, other algorithms were proposed. For ICA, the most notable ones are probably the FastICA algorithm [20] and the JADE algorithm [21]. The FastICA algorithm, with appropriate choice of a contrast function, estimates iteratively the topics, making use of the orthonormal structure (17), and performs the deflation procedure at every step. The recently introduced tensor power method (TPM) for the LDA model [4] is close to the FastICA algorithm. Alternatively, the JADE algorithm modifies the spectral algorithm by performing *multiple* projections for (17) and then jointly diagonalizing the resulting matrices with an orthogonal matrix. The spectral algorithm is a special case of this orthogonal joint diagonalization algorithm when only one projection is chosen. Importantly, a fast implementation [23] of the orthogonal joint diagonalization algorithm from [24] was proposed, which is based on closed-form iterative Jacobi updates (see, e.g., [25] for the later).

In practice, the orthogonal joint diagonalization (JD) algorithm is more robust than FastICA (see, e.g., [26, p. 30]) or the spectral algorithm. Moreover, although the application of the JD algorithm for the learning of topic models was mentioned in the literature [4, 27], it was never implemented in practice. In this paper, we apply the JD algorithm for the diagonalization of the GP/DICA cumulants as well as the LDA moments, which is described in Algorithm 1. Note that the choice of a projection vector $v_p \in \mathbb{R}^M$ obtained as $v_p = \widehat{W}^\top u_p$ for some vector $u_p \in \mathbb{R}^K$ is important and corresponds to the multilinear transformation of $\widehat{T}$ with $\widehat{W}^\top$ along the third mode. Importantly, in Algorithm 1, the joint diagonalization routine is performed over $(P+1)$ matrices of size $K \times K$, where the number of topics $K$ is usually not too big. This makes the algorithm computationally fast (see Appendix G.1). The same is true for the spectral algorithm, but not for TPM.

In Section 5.1, we compare experimentally the performance of the spectral, JD, and TPM algorithms for the estimation of the parameters of the GP/DICA as well as LDA models. We are not aware of any experimental comparison of these algorithms in the LDA context. While already working on this manuscript, the JD algorithm was also independently analyzed by [27] in the context of tensor factorization for general latent variable models. However, [27] focused mostly on the comparison of approaches for tensor factorization and their stability properties, with brief experiments using a latent variable model related but not equivalent to LDA for community detection. In contrast, we provide a detailed experimental comparison in the context of LDA in this paper, as well as propose a novel cumulant-based estimator. Due to the space restriction the estimation of the topic matrix $D$ and the (gamma/Dirichlet) parameter $c$ are moved to Appendix F.6.

**Algorithm 1** Joint diagonalization (JD) algorithm for GP/DICA cumulants (or LDA moments)

---

1: *Input:* $X \in \mathbb{R}^{M \times N}$, $K$, $P$ (number of random projections); (and $c_0$ for LDA moments)

2: Compute sample estimate $\widehat{S} \in \mathbb{R}^{M \times M}$ ((29) for GP/DICA / (43) for LDA in Appendix F)

3: Estimate whitening matrix $\widehat{W} \in \mathbb{R}^{K \times M}$ of $\widehat{S}$ (see Appendix F.1)
   *option (a):* Choose vectors $\{u_1, u_2, \ldots, u_P\} \subseteq \mathbb{R}^K$ uniformly at random from the unit $\ell_2$-sphere and set $v_p = \widehat{W}^\top u_p \in \mathbb{R}^M$ for all $p = 1, \ldots, P$      ($P = 1$ yields the spectral algorithm)
   *option (b):* Choose vectors $\{u_1, u_2, \ldots, u_P\} \subseteq \mathbb{R}^K$ as the canonical basis $e_1, e_2, \ldots, e_K$ of $\mathbb{R}^K$ and set $v_p = \widehat{W}^\top u_p \in \mathbb{R}^M$ for all $p = 1, \ldots, K$

4: For $\forall p$, compute $B_p = \widehat{W}\widehat{T}(v_p)\widehat{W}^\top \in \mathbb{R}^{K \times K}$ ((52) for GP/DICA / (54) for LDA; Appendix F)

5: Perform orthogonal joint diagonalization of matrices $\{\widehat{W}\widehat{S}\widehat{W}^\top = I_K, \ B_p, \ p = 1, \ldots, P\}$ (see [24] and [23]) to find an orthogonal matrix $V \in \mathbb{R}^{K \times K}$ and vectors $\{a_1, a_2, \ldots, a_P\} \subset \mathbb{R}^K$ such that
$$V\widehat{W}\widehat{S}\widehat{W}^\top V^\top = I_K, \text{ and } V B_p V^\top \approx \text{diag}(a_p), \ p = 1, \ldots, P$$

6: Estimate joint diagonalization matrix $A = V\widehat{W}$ and values $a_p, p = 1, \ldots, P$

7: *Output:* Estimate of $D$ and $c$ as described in Appendix F.6

---

## 5 Experiments

In this section, (a) we compare experimentally the GP/DICA cumulants with the LDA moments and (b) the spectral algorithm [3], the tensor power method [4] (TPM), the joint diagonalization (JD) algorithm from Algorithm 1, and variational inference for LDA [1].

**Real data:** the associated press (AP) dataset, from D. Blei's web page,[4] with $N = 2,243$ documents and $M = 10,473$ vocabulary words and the average document length $\widehat{L} = 194$; the NIPS papers dataset[5] [28] of $2,483$ NIPS papers and $14,036$ words, and $\widehat{L} = 1,321$; the KOS dataset,[6] from the UCI Repository, with $3,430$ documents and $6,906$ words, and $\widehat{L} = 136$.

**Semi-synthetic data** are constructed by analogy with [29]: (1) the LDA parameters $D$ and $c$ are learned from the real datasets with variational inference and (2) toy data are sampled from a model of interest with the given parameters $D$ and $c$. This provides the ground truth parameters $D$ and $c$. For each setting, data are sampled 5 times and the results are averaged. We plot error bars that are the minimum and maximum values. For the AP data, $K \in \{10, 50\}$ topics are learned and, for the NIPS data, $K \in \{10, 90\}$ topics are learned. For larger $K$, the obtained topic matrix is ill-conditioned, which violates the identifiability condition for topic recovery using moment matching techniques [3]. All the documents with less than 3 tokens are resampled.

**Sampling techniques.** All the sampling models have the parameter $c$ which is set to $c = c_0 \bar{c} / \|\bar{c}\|_1$, where $\bar{c}$ is the learned $c$ from the real dataset with variational LDA, and $c_0$ is a parameter that we can vary. The *GP* data are sampled from the gamma-Poisson model (3) with $b = c_0/\widehat{L}$ so that the expected document length is $\widehat{L}$ (see Appendix B.2). The *LDA-fix(L)* data are sampled from the LDA model (2) with the document length being fixed to a given $L$. The *LDA-fix2($\gamma$,$L_1$,$L_2$)* data are sampled as follows: $(1 - \gamma)$-portion of the documents are sampled from the *LDA-fix($L_1$)* model with a given document length $L_1$ and $\gamma$-portion of the documents are sampled from the *LDA-fix($L_2$)* model with a given document length $L_2$.

**Evaluation.** Evaluation of topic recovery for semi-synthetic data is performed with the $\ell_1$-error between the recovered $\widehat{D}$ and true $D$ topic matrices with the best permutation of columns: $\text{err}_{\ell_1}(\widehat{D}, D) := \min_{\pi \in \text{PERM}} \frac{1}{2K} \sum_k \|\widehat{d}_{\pi_k} - d_k\|_1 \in [0, 1]$. The minimization is over the possible permutations $\pi \in \text{PERM}$ of the columns of $\widehat{D}$ and can be efficiently obtained with the Hungarian algorithm for bipartite matching. For the evaluation of topic recovery in the real data case, we use an approximation of the log-likelihood for held out documents as the metric [16]. See Appendix G.6 for more details.

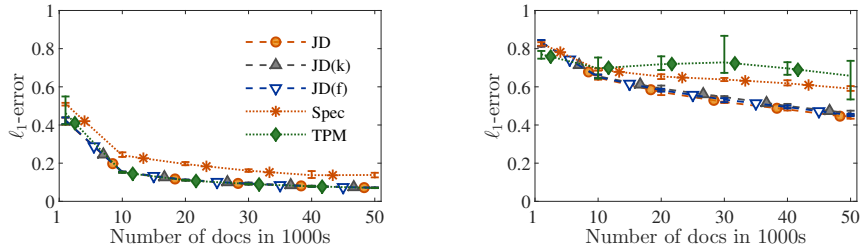

Figure 1: Comparison of the diagonalization algorithms. The topic matrix $D$ and Dirichlet parameter $c$ are learned for $K = 50$ from AP; $c$ is scaled to sum up to $0.5$ and $b$ is set to fit the expected document length $\widehat{L} = 200$. The semi-synthetic dataset is sampled from *GP*; number of documents $N$ varies from $1,000$ to $50,000$. **Left:** GP/DICA moments. **Right:** LDA moments. *Note*: a smaller value of the $\ell_1$-error is better.

We use our Matlab implementation of the GP/DICA cumulants, the LDA moments, and the diagonalization algorithms. The datasets and the code for reproducing our experiments are available online.[7] In Appendix G.1, we discuss implementation and complexity of the algorithms. We explain how we initialize the parameter $c_0$ for the LDA moments in Appendix G.3.

### 5.1 Comparison of the diagonalization algorithms

In Figure 1, we compare the diagonalization algorithms on the semi-synthetic AP dataset for $K = 50$ using the GP sampling. We compare the tensor power method (TPM) [4], the spectral algorithm (Spec), the orthogonal joint diagonalization algorithm (JD) described in Algorithm 1 with different options to choose the random projections: JD(k) takes $P = K$ vectors $u_p$ sampled uniformly from the unit $\ell_2$-sphere in $\mathbb{R}^K$ and selects $v_p = W^\top u_p$ (option (a) in Algorithm 1); JD selects the full basis $e_1, \ldots, e_K$ in $\mathbb{R}^K$ and sets $v_p = W^\top e_p$ (as JADE [21]) (option (b) in Algorithm 1); $JD(f)$ chooses the full canonical basis of $\mathbb{R}^M$ as the projection vectors (computationally expensive).

Both the GP/DICA cumulants and LDA moments are well-specified in this setup. However, the LDA moments have a slower finite sample convergence and, hence, a larger estimation error for the same value $N$. As expected, the spectral algorithm is always slightly inferior to the joint diagonalization algorithms. With the GP/DICA cumulants, where the estimation error is low, all algorithms demonstrate good performance, which also fulfills our expectations. However, although TPM shows almost perfect performance in the case of the GP/DICA cumulants (left), it significantly deteriorates for the LDA moments (right), which can be explained by the larger estimation error of the LDA moments and lack of robustness of TPM. The running times are discussed in Appendix G.2. Overall, the orthogonal joint diagonalization algorithm with initialization of random projections as $W^\top$ multiplied with the canonical basis in $\mathbb{R}^K$ (JD) is both computationally efficient and fast.

### 5.2 Comparison of the GP/DICA cumulants and the LDA moments

In Figure 2, when sampling from the *GP* model (top, left), both the GP/DICA cumulants and LDA moments are well specified, which implies that the approximation error (i.e., the error for the infinite number of documents) is low for both. The GP/DICA cumulants achieve low values of the estimation error already for $N = 10,000$ documents independently of the number of topics, while the convergence is slower for the LDA moments. When sampling from the *LDA-fix(200)* model (top, right), the GP/DICA cumulants are mis-specified and their approximation error is high, although the estimation error is low due to the faster finite sample convergence. One reason of poor performance of the GP/DICA cumulants, in this case, is the absence of variance in document length. Indeed, if documents with two different lengths are mixed by sampling from the *LDA-fix2(0.5,20,200)* model (bottom, left), the GP/DICA cumulants performance improves. Moreover, the experiment with a changing fraction $\gamma$ of documents (bottom, right) shows that a non-zero variance on the length improves the performance of the GP/DICA cumulants. As in practice real corpora usually have a non-zero variance for the document length, this bad scenario for the GP/DICA cumulants is not likely to happen.

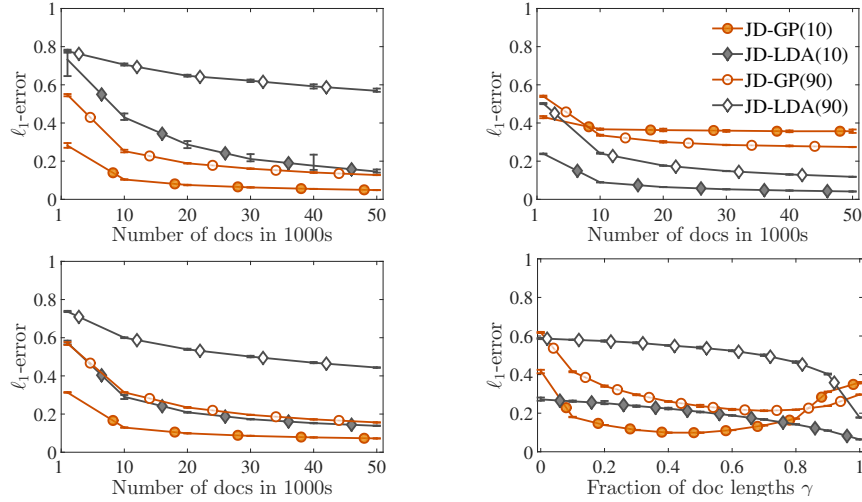

Figure 2: Comparison of the GP/DICA cumulants and LDA moments. Two topic matrices and parameters $c_1$ and $c_2$ are learned from the NIPS dataset for $K = 10$ and 90; $c_1$ and $c_2$ are scaled to sum up to $c_0 = 1$. Four corpora of different sizes $N$ from $1,000$ to $50,000$: **top, left:** $b$ is set to fit the expected document length $\widehat{L} = 1300$; sampling from the *GP* model; **top, right:** sampling from the *LDA-fix(200)* model; **bottom, left:** sampling from the *LDA-fix2(0.5,20,200)* model. **Bottom, right:** the number of documents here is fixed to $N = 20,000$; sampling from the *LDA-fix2($\gamma$,20,200)* model varying the values of the fraction $\gamma$ from 0 to 1 with the step 0.1. *Note*: a smaller value of the $\ell_1$-error is better.

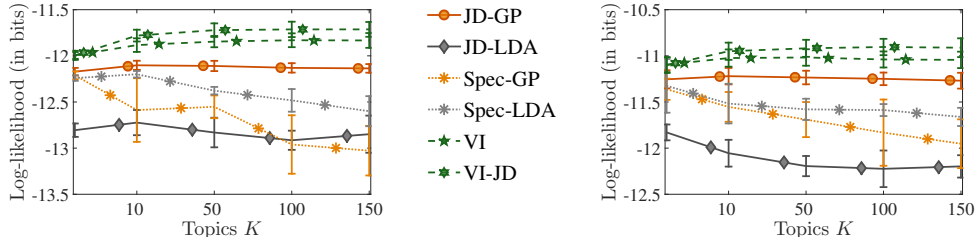

Figure 3: Experiments with real data. **Left:** the AP dataset. **Right:** the KOS dataset. *Note*: a higher value of the log-likelihood is better.

## 5.3 Real data experiments

In Figure 3, JD-GP, Spec-GP, JD-LDA, and Spec-LDA are compared with variational inference (VI) and with variational inference initialized with the output of JD-GP (VI-JD). We measure held out log-likelihood per token (see Appendix G.7 for details on the experimental setup). The orthogonal joint diagonalization algorithm with the GP/DICA cumulants (JD-GP) demonstrates promising performance. In particular, the GP/DICA cumulants significantly outperform the LDA moments. Moreover, although variational inference performs better than the JD-GP algorithm, restarting variational inference with the output of the JD-GP algorithm systematically leads to better results. Similar behavior has already been observed (see, e.g., [30]).

## 6 Conclusion

In this paper, we have proposed a new set of tensors for a discrete ICA model related to LDA, where word counts are directly modeled. These moments make fewer assumptions regarding distributions, and are theoretically and empirically more robust than previously proposed tensors for LDA, both on synthetic and real data. Following the ICA literature, we showed that our joint diagonalization procedure is also more robust. Once the topic matrix has been estimated in a semi-parametric way where topic intensities are left unspecified, it would be interesting to learn the unknown distributions of the independent topic intensities.

**Acknowledgments.** This work was partially supported by the MSR-Inria Joint Center. The authors would like to thank Christophe Dupuy for helpful discussions.

## Footnotes

[1]Note that the expected squared error for the DICA cumulants is similar, but the expressions are less compact and, in general, depend on the prior on $\alpha_k$.

[2]For completeness, we also present the finite sample estimates $\widehat{S}$ (43) and $\widehat{T}$ (44) of $S$ (39) and $T$ (40) for the LDA moments (which are consistent with the ones suggested in [4]) in Appendix F.4.

[3]See Appendix C.5 for a discussion on the orders.

[4] http://www.cs.columbia.edu/~blei/lda-c

[5] http://ai.stanford.edu/~gal/data

[6] https://archive.ics.uci.edu/ml/datasets/Bag+of+Words

[7] https://github.com/anastasia-podosinnikova/dica

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
