[Supplementary Material · nips_full_supplementary.pdf]

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

[8]Strictly speaking, the (scalar) $n$-th cumulant $\kappa_n$ of a random variable $X$ is defined via the cumulant-generating function $g(t)$, which is the natural logarithm of the moment-generating function, i.e $g(t) := \log \mathbb{E}\left[e^{tX}\right]$. The cumulant $\kappa_n$ is then obtained from a power series expansion of the cumulant-generating function, that is $g(t) = \sum_{n=1}^\infty \kappa_n t^n / n!$ [Wikipedia].

[9]In [4], given a tensor $T \in \mathbb{R}^{K \times K \times K}$, $T(D^\top, D^\top, D^\top)$ is referred to as the multilinear map. In [34], the same entity is denoted by $T \times_1 D^\top \times_2 D^\top \times_3 D^\top$, where $\times_n$ denotes the $n$-mode tensor-matrix product.

[10]For tensors, such decomposition is also known under the names CANDECOMP/PARAFAC or, simply, the CP decomposition (see, e.g., [34]).

[11]Note that another advantage of the DICA cumulants from Section 3.1 is that they do not require such a somewhat artificial condition: they are well-defined for any document length (even a document of length zero!).

[12]Note, the difference in the notation for the LDA moments in papers [3] and [4]. In [3], $M_1 = \mathbb{E}(w_{\ell_1})$, $M_2 = \mathbb{E}(w_{\ell_1} \otimes w_{\ell_2})$, and $M_3 = \mathbb{E}(w_{\ell_1} \otimes w_{\ell_2} \otimes w_{\ell_3})$. However, in [4], $M_2$ is equivalent to $S$ in our notation and to $Pairs$ in the notation of [3]; similarly, $M_3$ is $T$ in our notation or $Triples$ in the notation of [3].

[13]Note that because non-linear functions of $\widehat{M}_1$ appear in the expression for $\widehat{S}$ (43) and $\widehat{T}$ (44), the estimator is biased, i.e., $\mathbb{E}(\widehat{S}) \neq S$. The bias is small though: $\|\mathbb{E}(\widehat{S}) - S\| = O(1/N)$ and the estimator is asymptotically unbiased. This is in contrast with the estimator for the GP/DICA moments which is easily made unbiased.

[14]Note that such a whitening matrix $W \in \mathbb{R}^{K \times M}$ is not uniquely defined as left multiplication by any orthogonal matrix $V \in \mathbb{R}^{K \times K}$ does not change anything. Indeed, let $\widetilde{W} = VW$, then $\widetilde{W}S\widetilde{W}^\top = VWSW^\top V^\top = I_K$.

[15]We mean the largest non-negative eigenvalues. In theory, $S$ have to be PSD. In practice, when we deal with finite number of samples, respective estimate of $S$ can have negative eigenvalues. However, for $K$ sufficiently small, $S$ should have enough positive eigenvalues. Moreover, it is standard practice to use eigenvalues of $S$ for estimation of a good value of $K$, e.g., by thresholding all negative and close to zero eigenvalues.

[16] https://github.com/anastasia-podosinnikova/dica-light

[17] https://github.com/anastasia-podosinnikova/dica

[18] http://homepages.inf.ed.ac.uk/imurray2/pub/09etm

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

## A    Appendix. Plate diagrams for the models from Section 2

Figure 4: Plate diagrams for the models from Section 2.

In Section 2, index $n$, which stands for the $n$-th document, was omitted. For convenience, we recall the models: the LDA model in the tokens representation:

$$\begin{aligned}
\theta_n &\sim \text{Dirichlet}(c), \\
z_{n\ell}|\theta_n &\sim \text{Multinomial}(1, \theta_n), \\
w_{n\ell}|z_{n\ell}, \theta_n &\sim \text{Multinomial}(1, d_{z_{n\ell}});
\end{aligned} \tag{18}$$

the LDA model with marginalized out latent variable $z$:

$$\begin{aligned}
\theta_n &\sim \text{Dirichlet}(c), \\
x_n|\theta_n &\sim \text{Multinomial}(L_n, D\theta_n);
\end{aligned} \tag{19}$$

the GP model:

$$\begin{aligned}
\alpha_{nk} &\sim \text{Gamma}(c_k, b), \\
x_{nm}|\alpha_n &\sim \text{Poisson}([D\alpha_n]_m);
\end{aligned} \tag{20}$$

and the DICA model:

$$\begin{aligned}
\alpha_{n1}, \ldots, \alpha_{nK} &\sim \text{mutually independent}, \\
x_{nm}|\alpha_n &\sim \text{Poisson}([D\alpha_n]_m).
\end{aligned} \tag{21}$$

## B    Appendix. The GP model

### B.1    The connection between the LDA and GP models

To show that the LDA model (2) with the additional assumption that the document length is modeled as a gamma-Poisson random variable is equivalent to the GP model (3), we show that:

- when modeling the document length $L$ as a Poisson random variable with parameter $\lambda$, the count vectors $x_1, x_2, \ldots, x_M$ are mutually independent Poisson random variables;

- the Gamma prior on $\lambda$ reveals the connection $\alpha_k = \lambda \theta_k$ between the Dirichlet random variable $\theta$ and the mutually independent gamma random variables $\alpha_1, \alpha_2, \ldots, \alpha_K$.

For completeness, we repeat the known result that if $L \sim \text{Poisson}(\lambda)$ and $x|L \sim \text{Multinomial}(L, D\theta)$ (which thus means that $L = \sum_m x_m$ with probability one), then $x_1, x_2, \ldots, x_M$ are mutually independent Poisson random variables with parameters $\lambda[D\theta]_1, \lambda[D\theta]_2, \ldots,$

$\lambda \left[ D\theta \right]_M$. Indeed, we consider the following joint probability mass function where $x$ and $L$ are assumed to be non-negative integers:

$$
\begin{aligned}
p(x, L|\theta, \lambda) =& p(L|\lambda)p(x|L, \theta) \\
=& \mathbb{1}_{\{L = \sum_m x_m\}} \frac{\exp{(-\lambda)} \lambda^L}{\cancel{L!}} \frac{\cancel{L!}}{\prod_m x_m!} \prod_m [D\theta]_m^{x_m} \\
=& \mathbb{1}_{\{L = \sum_m x_m\}} \exp(-\lambda \sum_m [D\theta]_m) \lambda^{\sum_m x_m} \prod_m \frac{[D\theta]_m^{x_m}}{x_m!} \\
=& \mathbb{1}_{\{L = \sum_m x_m\}} \prod_m \frac{\exp(-\lambda [D\theta]_m)(\lambda [D\theta]_m)^{x_m}}{x_m!} \\
=& \mathbb{1}_{\{L = \sum_m x_m\}} \prod_m \mathrm{Poisson}(x_m; \lambda [D\theta]_m),
\end{aligned}
$$

where in the third equation we used the fact that

$$
\sum_m [D\theta]_m = \sum_{m,k} D_{mk}\theta_k = \sum_k \theta_k \sum_m D_{mk} = 1.
$$

We thus have $p(x, L|\theta, \lambda) = p(L|x) \prod_m p(x_m|\lambda[D\theta]_m)$ where $p(L|x)$ is simply the deterministic distribution $\mathbb{1}_{\{L = \sum_m x_m\}}$ and $p(x_m|\lambda[D\theta]_m)$ for $m = 1, \dots, M$ are independent $\mathrm{Poisson}(\lambda[D\theta]_m)$ distributions (and thus do not depend on $L$). Note that in the notation introduced in the paper, $D_{mk} = d_{km}$. Hence, by using the construction of the Dirichlet distribution from the normalization of independent gamma random variables, we can show that the LDA model with a gamma-Poisson prior over the length is equivalent to the following model (recall, that $c_0 = \sum_k c_k$):

$$
\begin{aligned}
\lambda &\sim \mathrm{Gamma}(c_0, b), \\
\theta &\sim \mathrm{Dirichlet}(c), \\
x_m|\lambda, \theta &\sim \mathrm{Poisson}([D(\lambda\theta)]_m).
\end{aligned}
\tag{22}
$$

More specifically, we complete the second part of the argument with the following properties. When $\alpha_1, \alpha_2, \dots, \alpha_K$ are mutually independent gamma random variables, each $\alpha_k \sim \mathrm{Gamma}(c_k, b)$, their sum is also a gamma random variable $\sum_k \alpha_k \sim \mathrm{Gamma}(\sum_k c_k, b)$. The former is equivalent to $\lambda$. It is known (e.g., [32]) that a Dirichlet random variable can be sampled by first sampling independent gamma random variables ($\alpha_k$) and then dividing each of them by their sum ($\lambda$): $\theta_k = \alpha_k / \sum_{k'} \alpha_{k'}$, and, in other direction, the variables $\alpha_k = \lambda\theta_k$ are mutually independent, giving back the GP model (3).

## B.2 The expectation and the variance of the document length for the GP model

From the drivations in Appendix B.1, it follows that the document length of the GP model (3) is a gamma-Poisson random variable, i.e., $L|\lambda \sim \mathrm{Poisson}(\lambda)$ and $\lambda \sim \mathrm{Gamma}(c_0, b)$. Therefore, the following follows from the law of total expectation and the law of total variance

$$
\begin{aligned}
\mathbb{E}(L) &= \mathbb{E}\left[\mathbb{E}(L|\lambda)\right] = \mathbb{E}(\lambda) = c_0/b \\
\mathrm{var}(L) &= \mathrm{var}\left[\mathbb{E}(L|\lambda)\right] + \mathbb{E}\left[\mathrm{var}(L|\lambda)\right] = \mathrm{var}(\lambda) + \mathbb{E}(\lambda) = c_0/b + c_0/b^2
\end{aligned}
$$

The first expression shows that the parameter $b$ controls the expected document length $\mathbb{E}(L)$ for a given parameter $c_0$: the smaller $b$, the larger $\mathbb{E}(L)$. On the other hand, if we allow $c_0$ to vary as well, only the ratio $c_0/b$ is important for the document length. We can then interpret the role of $c_0$ as actually controlling the concentration of the distribution for the length $L$ (through the variance). More specifically, we have that:

$$
\frac{\mathrm{var}(L)}{(\mathbb{E}(L))^2} = \frac{1}{\mathbb{E}(L)} + \frac{1}{c_0}.
\tag{23}
$$

For a fixed target document length $\mathbb{E}(L)$, we can increase the variance (and thus decrease the concentration) by using a smaller $c_0$.

## C  Appendix. The cumulants of the GP and DICA models

### C.1  Cumulants

For a random vector $x \in \mathbb{R}^M$, the first three cumulant tensors[8] are

$$\mathrm{cum}(x_m) = \mathbb{E}(x_m),$$
$$\mathrm{cum}(x_{m_1}, x_{m_2}) = \mathbb{E}\left[(x_{m_1} - \mathbb{E}(x_{m_1}))(x_{m_2} - \mathbb{E}(x_{m_2}))\right] = \mathrm{cov}(x_{m_1}, x_{m_2}),$$
$$\mathrm{cum}(x_{m_1}, x_{m_2}, x_{m_3}) = \mathbb{E}\left[(x_{m_1} - \mathbb{E}(x_{m_1}))(x_{m_2} - \mathbb{E}(x_{m_2}))(x_{m_3} - \mathbb{E}(x_{m_3})))\right].$$

Note that the 2nd and 3rd cumulants coincide with the 2nd and 3rd central moments (but not for higher orders). In the following, $\mathrm{cum}(x, x, x) \in \mathbb{R}^{M \times M \times M}$ denotes the third order tensor with elements $\mathrm{cum}(x_{m_1}, x_{m_2}, x_{m_3})$. Some of the properties of cumulants are listed below (see [5, chap. 5]). The most important property that motivate us to use cumulants in this paper (and the ICA literature) is the **independence** property, which says that the cumulant tensor for a random vector with independent components is diagonal (this property *does not* hold for the (non-central) moment tensors of any order, and neither for the central moments of order 4 or more).

- **Independence.** If the elements of $x \in \mathbb{R}^M$ are independent, then their cross-cumulants are zero as soon as two indices are different, i.e., $\mathrm{cum}(x_{m_1}, x_{m_2}) = \delta(m_1, m_2)\mathbb{E}[(x_{m_1} - \mathbb{E}_{m_1}))^2]$ and $\mathrm{cum}(x_{m_1}, x_{m_2}, x_{m_3}) = \delta(m_1, m_2, m_3)\mathbb{E}[(x_{m_1} - \mathbb{E}(x_{m_1}))^3]$, where $\delta$ is the Kronecker delta.

- **Multilinearity.** If two random vectors $y \in \mathbb{R}^M$ and $\alpha \in \mathbb{R}^K$ are linearly dependent, i.e., $y = D\alpha$ for some $D \in \mathbb{R}^{M \times K}$, then

$$\mathrm{cum}(y_m) = \sum_k \mathrm{cum}(\alpha_k)D_{mk},$$
$$\mathrm{cum}(y_{m_1}, y_{m_2}) = \sum_{k_1, k_2} \mathrm{cum}(\alpha_{k_1}, \alpha_{k_2})D_{m_1 k_1}D_{m_2 k_2},$$
$$\mathrm{cum}(y_{m_1}, y_{m_2}, y_{m_3}) = \sum_{k_1, k_2, k_3} \mathrm{cum}(\alpha_{k_1}, \alpha_{k_2}, \alpha_{k_3})D_{m_1 k_1}D_{m_2 k_2}D_{m_3 k_3},$$

which can also be denoted[9] by

$$\mathbb{E}(y) = D\mathbb{E}(\alpha),$$
$$\mathrm{cov}(y, y) = D\mathrm{cov}(\alpha, \alpha)D^\top,$$
$$\mathrm{cum}(y, y, y) = \mathrm{cum}(\alpha, \alpha, \alpha)(D^\top, D^\top, D^\top).$$

- **The law of total cumulance.** For two random vectors $x \in \mathbb{R}^M$ and $y \in \mathbb{R}^M$, it holds

$$\mathrm{cum}(x_m) = \mathbb{E}\left[\mathbb{E}(x_m|y)\right],$$
$$\mathrm{cum}(x_{m_1}, x_{m_2}) = \mathbb{E}\left[\mathrm{cov}(x_{m_1}, x_{m_2}|y)\right] + \mathrm{cov}\left[\mathbb{E}(x_{m_1}|y), \mathbb{E}(x_{m_2}|y)\right],$$
$$\mathrm{cum}(x_{m_1}, x_{m_2}, x_{m_3}) = \mathbb{E}\left[\mathrm{cum}(x_{m_1}, x_{m_2}, x_{m_3}|y)\right] + \mathrm{cum}\left[\mathbb{E}(x_{m_1}|y), \mathbb{E}(x_{m_2}|y), \mathbb{E}(x_{m_3}|y)\right]$$
$$+ \mathrm{cov}\left[\mathbb{E}(x_{m_1}|y), \mathrm{cov}(x_{m_2}, x_{m_3}|y)\right]$$
$$+ \mathrm{cov}\left[\mathbb{E}(x_{m_2}|y), \mathrm{cov}(x_{m_1}, x_{m_3}|y)\right]$$
$$+ \mathrm{cov}\left[\mathbb{E}(x_{m_3}|y), \mathrm{cov}(x_{m_1}, x_{m_2}|y)\right].$$

Note that the first expression is also well known as the law of total expectation or the tower property, while the second one is known as the law of total covariance.

## C.2 The third cumulant of the GP/DICA models

In this section, by analogy with Section 3.1, we derive the third GP/DICA cumulant.

As the third cumulant of a Poisson random variable $x_m$ with parameter $y_m$ is $\mathbb{E}((x_m - \mathbb{E}(x_m))^3|y_m) = y_m$, then by the independence property of cumulants from Section C.1, the cumulant of $x|y$ is diagonal:

$$\mathrm{cum}(x_{m_1}, x_{m_2}, x_{m_3}|y) = \delta(m_1, m_2, m_3)\, y_{m_1}.$$

Substituting the cumulant of $x|y$ into the law of total cumulance, we obtain

$$
\begin{aligned}
\mathrm{cum}(x_{m_1}, x_{m_2}, x_{m_3}) &= \mathbb{E}\left[\mathrm{cum}(x_{m_1}, x_{m_2}, x_{m_3}|y)\right] \\
&\quad + \mathrm{cum}\left[\mathbb{E}(x_{m_1}|y), \mathbb{E}(x_{m_2}|y), \mathbb{E}(x_{m_3}|y)\right] + \mathrm{cov}\left[\mathbb{E}(x_{m_1}|y), \mathrm{cov}(x_{m_2}, x_{m_3}|y)\right] \\
&\quad + \mathrm{cov}\left[\mathbb{E}(x_{m_2}|y), \mathrm{cov}(x_{m_1}, x_{m_3}|y)\right] + \mathrm{cov}\left[\mathbb{E}(x_{m_3}|y), \mathrm{cov}(x_{m_1}, x_{m_2}|y)\right] \\
&= \delta(m_1, m_2, m_3)\mathbb{E}(y_{m_1}) + \mathrm{cum}(y_{m_1}, y_{m_2}, y_{m_3}) \\
&\quad + \delta(m_2, m_3)\mathrm{cov}(y_{m_1}, y_{m_2}) + \delta(m_1, m_3)\mathrm{cov}(y_{m_1}, y_{m_2}) + \delta(m_1, m_2)\mathrm{cov}(y_{m_1}, y_{m_3}) \\
&= \delta(m_1, m_2, m_3)\mathbb{E}(x_{m_1}) + \mathrm{cum}(y_{m_1}, y_{m_2}, y_{m_3}) \\
&\quad + \delta(m_2, m_3)\mathrm{cov}(x_{m_1}, x_{m_2}) - \delta(m_1, m_2, m_3)\mathbb{E}(x_{m_1}) \\
&\quad + \delta(m_1, m_3)\mathrm{cov}(x_{m_1}, x_{m_2}) - \delta(m_1, m_2, m_3)\mathbb{E}(x_{m_1}) \\
&\quad + \delta(m_1, m_2)\mathrm{cov}(x_{m_1}, x_{m_3}) - \delta(m_1, m_2, m_3)\mathbb{E}(x_{m_1}) \\
&= \mathrm{cum}(y_{m_1}, y_{m_2}, y_{m_3}) - 2\delta(m_1, m_2, m_3)\mathbb{E}(x_{m_1}) \\
&\quad + \delta(m_2, m_3)\mathrm{cov}(x_{m_1}, x_{m_2}) + \delta(m_1, m_3)\mathrm{cov}(x_{m_1}, x_{m_2}) + \delta(m_1, m_2)\mathrm{cov}(x_{m_1}, x_{m_3}) \\
&= \left[\mathrm{cum}(\alpha, \alpha, \alpha)(D^\top, D^\top, D^\top)\right]_{m_1 m_2 m_3} - 2\delta(m_1, m_2, m_3)\mathbb{E}(x_{m_1}) \\
&\quad + \delta(m_2, m_3)\mathrm{cov}(x_{m_1}, x_{m_2}) + \delta(m_1, m_3)\mathrm{cov}(x_{m_1}, x_{m_2}) + \delta(m_1, m_2)\mathrm{cov}(x_{m_1}, x_{m_3}),
\end{aligned}
\tag{24}
$$

where, in the third equality, we used the previous result from (9) that $\mathrm{cov}(y, y) = \mathrm{cov}(x, x) - \mathrm{diag}(\mathbb{E}(x))$.

## C.3 The diagonal structure of the GP/DICA cumulants

In this section, we provide detailed derivation of the diagonal structure (11) of the matrix $S$ (10) and the diagonal structure (13) of the tensor $T$ (12).

From the independence of $\alpha_1, \alpha_2, \ldots, \alpha_K$ and by the independence property of cumulants from Section C.1, it follows that $\mathrm{cov}(\alpha, \alpha)$ is a diagonal matrix and $\mathrm{cum}(\alpha, \alpha, \alpha)$ is a diagonal tensor, i.e., $\mathrm{cov}(\alpha_{k_1}, \alpha_{k_2}) = \delta(k_1, k_2)\mathrm{cov}(\alpha_{k_1}, \alpha_{k_2})$ and $\mathrm{cum}(\alpha_{k_1}, \alpha_{k_2}, \alpha_{k_3}) = \delta(k_1, k_2, k_3)\mathrm{cum}(\alpha_{k_1}, \alpha_{k_1}, \alpha_{k_1})$. Therefore, the following holds

$$\mathrm{cov}(y_{m_1}, y_{m_2}) = \sum_k \mathrm{cov}(\alpha_k, \alpha_k)D_{m_1 k}D_{m_2 k},$$

$$\mathrm{cum}(y_{m_1}, y_{m_2}, y_{m_3}) = \sum_k \mathrm{cum}(\alpha_k, \alpha_k, \alpha_k)D_{m_1 k}D_{m_2 k}D_{m_3 k},$$

which we can rewrite in a matrix/tensor form as

$$\mathrm{cov}(y, y) = \sum_k \mathrm{cov}(\alpha_k, \alpha_k)d_k d_k^\top,$$

$$\mathrm{cum}(y, y, y) = \sum_k \mathrm{cum}(\alpha_k, \alpha_k, \alpha_k)d_k \otimes d_k \otimes d_k.$$

Moving $\mathrm{cov}(y, y)\,/\,\mathrm{cum}(y, y, y)$ in the expression for $\mathrm{cov}(x, x)$ (9)$\,/\,\mathrm{cum}(x, x, x)$ (24) on one side of equality and all other terms on the other side, we define matrix $S \in \mathbb{R}^{M \times M}\,/$ tensor $T \in \mathbb{R}^{M \times M \times M}$ as follows

$$S := \mathrm{cov}(x, x) - \mathrm{diag}\left(\mathbb{E}(x)\right), \tag{25}$$

$$
\begin{aligned}
T_{m_1 m_2 m_3} &:= \mathrm{cum}(x_{m_1}, x_{m_2}, x_{m_3}) + 2\delta(m_1, m_2, m_3)\mathbb{E}(x_{m_1}) \\
&\quad - \delta(m_2, m_3)\mathrm{cov}(x_{m_1}, x_{m_2}) \\
&\quad - \delta(m_1, m_3)\mathrm{cov}(x_{m_1}, x_{m_2}) \\
&\quad - \delta(m_1, m_2)\mathrm{cov}(x_{m_1}, x_{m_3}).
\end{aligned}
\tag{26}
$$

By construction, $S = \mathrm{cov}(y, y)$ and $T = \mathrm{cum}(y, y, y)$ and, therefore, it holds that

$$S = \sum_k \mathrm{cov}(\alpha_k, \alpha_k) d_k d_k^\top, \tag{27}$$

$$T = \sum_k \mathrm{cum}(\alpha_k, \alpha_k, \alpha_k) d_k \otimes d_k \otimes d_k. \tag{28}$$

This means that both the matrix $S$ and the tensor $T$ are sums of rank-1 matrices and tensors, respectively[10]. This structure of the matrix $S$ and the tensor $T$ is the basis for the algorithms considered in this paper.

### C.4 Unbiased finite sample estimators for the GP/DICA cumulants

Given a sample $\{x_1, x_2, \ldots, x_N\}$, we obtain a finite sample estimate $\widehat{S}$ of $S$ (10) / $\widehat{T}$ of $T$ (12) for the GP/DICA cumulants:

$$\widehat{S} := \widehat{\mathrm{cov}}(x, x) - \mathrm{diag}\left(\widehat{\mathbb{E}}(x)\right), \tag{29}$$

$$\begin{aligned}
\widehat{T}_{m_1 m_2 m_3} := {}& \widehat{\mathrm{cum}}(x_{m_1}, x_{m_2}, x_{m_3}) + 2\delta(m_1, m_2, m_3)\widehat{\mathbb{E}}(x_{m_1}) \\
& - \delta(m_2, m_3)\widehat{\mathrm{cov}}(x_{m_1}, x_{m_2}) \\
& - \delta(m_1, m_3)\widehat{\mathrm{cov}}(x_{m_1}, x_{m_2}) \\
& - \delta(m_1, m_2)\widehat{\mathrm{cov}}(x_{m_1}, x_{m_3}),
\end{aligned} \tag{30}$$

where unbiased estimators of the first three cumulants are

$$\widehat{\mathbb{E}}(x_{m_1}) = \frac{1}{N} \sum_n x_{nm_1},$$

$$\widehat{\mathrm{cov}}(x_{m_1}, x_{m_2}) = \frac{1}{N-1} \sum_n z_{nm_1} z_{nm_2}, \tag{31}$$

$$\widehat{\mathrm{cum}}(x_{m_1}, x_{m_2}, x_{m_3}) = \frac{N}{(N-1)(N-2)} \sum_n z_{nm_1} z_{nm_2} z_{nm_3},$$

where the word vocabulary indexes are $m_1, m_2, m_3 = 1, 2, \ldots, M$ and the centered documents $z_{nm} := x_{nm} - \widehat{\mathbb{E}}(x_m)$. (The latter is introduced only for compact representation of (31) and is different from $z$ in the LDA model.)

### C.5 On the orders of cumulants

Note that the factorization of $S = \widetilde{D}\widetilde{D}^\top$ does not uniquely determine $\widetilde{D}$ as one can equivalently use $S = (\widetilde{D}U)(\widetilde{D}U)^\top$ with any orthogonal $K \times K$ matrix $U$. Therefore, one has to consider higher than the second order information. Moreover, in ICA the fourth-order tensors are used, because the third cumulant of the Gaussian distribution is zero, which is not the case in the DICA/LDA models, where the third order information is sufficient.

## D Appendix. The sketch of the proof for Proposition 3.1

### D.1 Expected squared error for the sample expectation

The sample expectation is $\widehat{\mathbb{E}}(x) = \frac{1}{N} \sum_n x_n$ is an unbiased estimator of the expectation and:

$$\begin{aligned}
\mathbb{E}\left(\|\widehat{\mathbb{E}}(x) - \mathbb{E}(x)\|_2^2\right) &= \sum_m \mathbb{E}\left[\left(\widehat{\mathbb{E}}(x_m) - \mathbb{E}(x_m)\right)^2\right] \\
&= \frac{1}{N^2} \sum_m \left[\mathbb{E}\left(\sum_n (x_{nm} - \mathbb{E}(x_m))^2\right) + \mathbb{E}\left(\sum_n \sum_{n \neq n'} (x_{nm} - \mathbb{E}(x_m))(x_{n'm} - \mathbb{E}(x_m))\right)\right] \\
&= \frac{1}{N} \sum_m \mathbb{E}\left[(x_m - \mathbb{E}(x_m))^2\right] = \frac{1}{N} \sum_m \mathrm{var}(x_m).
\end{aligned}$$

Further, by the law of total variance:

$$\mathbb{E}\left(\|\widehat{\mathbb{E}}(x) - \mathbb{E}(x)\|_2^2\right) = \frac{1}{N}\sum_m \left[\mathbb{E}(\mathrm{var}(x_m|y)) + \mathrm{var}(\mathbb{E}(x_m|y))\right] = \frac{1}{N}\sum_m \left[\mathbb{E}(y_m) + \mathrm{var}(y_m)\right]$$

$$= \frac{1}{N}\left[\sum_k \mathbb{E}(\alpha_k) + \sum_k \langle d_k, d_k\rangle \mathrm{var}(\alpha_k)\right],$$

using the fact that $\sum_m D_{mk} = 1$ for any $k$.

## D.2 Expected squared error for the sample covariance

The following finite sample estimator of the covariance $\mathrm{cov}(x,x) = \mathbb{E}(xx^\top) - \mathbb{E}(x)\mathbb{E}(x)^\top$

$$\widehat{\mathrm{cov}}(x,x) = \frac{1}{N-1}\sum_n x_n x_n^\top - \widehat{\mathbb{E}}(x)\widehat{\mathbb{E}}(x)^\top = \frac{1}{N-1}\sum_n \left(x_n x_n^\top - \frac{1}{N^2}\sum_{n'}\sum_{n''} x_{n'} x_{n''}^\top\right)$$

$$= \frac{1}{N}\sum_n \left(x_n x_n^\top - \frac{1}{N-1}x_n \sum_{n'\neq n} x_{n'}^\top\right)$$

$$\tag{32}$$

is unbiased, i.e., $\mathbb{E}(\widehat{\mathrm{cov}}(x,x)) = \mathrm{cov}(x,x)$. Its squared error is

$$\mathbb{E}\left(\|\widehat{\mathrm{cov}}(x,x) - \mathrm{cov}(x,x)\|_F^2\right) = \sum_{m,m'}\mathbb{E}\left[(\widehat{\mathrm{cov}}(x_m,x_{m'}) - \mathbb{E}[\widehat{\mathrm{cov}}(x_m,x_{m'})])^2\right].$$

The $m,m'$-th element of the sum above is equal to

$$\frac{1}{N^2}\sum_{n,n'}\mathrm{cov}\left(x_{nm}x_{nm'} - \frac{1}{N-1}x_{nm}\sum_{n''\neq n}x_{n''m'} \quad , \quad x_{n'm}x_{n'm'} - \frac{1}{N-1}x_{n'm}\sum_{n'''\neq n'}x_{n'''m'}\right)$$

$$= \frac{1}{N^2}\sum_{n,n'}\mathrm{cov}\left(x_{nm}x_{nm'}, x_{n'm}x_{n'm'}\right) - \frac{2}{N^2(N-1)}\sum_{n,n'}\mathrm{cov}\left(x_{nm}\sum_{n''\neq n}x_{n''m'}, x_{n'm}x_{n'm'}\right)$$

$$+ \frac{1}{N^2(N-1)^2}\sum_{n,n'}\mathrm{cov}\left(x_{nm}\sum_{n''\neq n}x_{n''m'}, x_{n'm}\sum_{n'''\neq n'}x_{n'''m'}\right)$$

$$= \frac{1}{N^2}\sum_n \mathrm{cov}\left(x_{nm}x_{nm'}, x_{nm}x_{nm'}\right)$$

$$- \frac{2}{N^2(N-1)}\left[\sum_n\sum_{n''\neq n}\mathrm{cov}\left(x_{nm}x_{n''m'}, x_{nm}x_{nm'}\right) + \sum_n\sum_{n'\neq n}\mathrm{cov}\left(x_{nm}x_{n'm'}, x_{n'm}x_{n'm'}\right)\right]$$

$$+ \frac{1}{N^2(N-1)^2}\left[\sum_n\sum_{n''\neq n}\sum_{n'''\neq n}\mathrm{cov}\left(x_{nm}x_{n''m'}, x_{nm}x_{n'''m'}\right) + \sum_{n'}\sum_{n\neq n'}\sum_{n''\neq n}\mathrm{cov}\left(x_{nm}x_{n''m'}, x_{n'm}x_{nm'}\right)\right]$$

$$+ \frac{1}{N^2(N-1)^2}\left[\sum_{n'}\sum_{n\neq n'}\sum_{n'''\neq n'}\mathrm{cov}\left(x_{nm}x_{n'm'}, x_{n'm}x_{n'''m'}\right) + \sum_{n'}\sum_{n\neq n'}\sum_{n''\neq n}\mathrm{cov}\left(x_{nm}x_{n''m'}, x_{n'm}x_{n''m'}\right)\right],$$

where we used mutual independence of the observations $x_n$ in a sample $\{x_n\}_{n=1}^N$ to conclude that the covariance between two expressions involving only independent variables is zero. Fur-

ther:

$$\mathbb{E}\left(\|\widehat{\mathrm{cov}}(x,x)-\mathrm{cov}(x,x)\|_F^2\right) = \frac{1}{N^2}\sum_{m,m'} N\left(\mathbb{E}(x_m^2 x_{m'}^2) - [\mathbb{E}(x_m x_{m'})]^2\right)$$

$$-\frac{4}{N^2(N-1)}\sum_{m,m'} N(N-1)\left(\mathbb{E}(x_m^2 x_{m'})\mathbb{E}(x_{m'}) - \mathbb{E}(x_m x_{m'})\mathbb{E}(x_m)\mathbb{E}(x_{m'})\right)$$

$$+\frac{2}{N^2(N-1)^2}\sum_{m,m'} N(N-1)(N-2)\left(\mathbb{E}(x_m^2)\left[\mathbb{E}(x_{m'})\right]^2 - \left[\mathbb{E}(x_m)\right]^2\left[\mathbb{E}(x_{m'})\right]^2\right)$$

$$+\frac{2}{N^2(N-1)^2}\sum_{m,m'} N(N-1)(N-2)\left(\mathbb{E}(x_m x_{m'})\mathbb{E}(x_m)\mathbb{E}(x_{m'}) - \left[\mathbb{E}(x_m)\right]^2\left[\mathbb{E}(x_{m'})\right]^2\right) + O\left(\frac{1}{N^2}\right),$$

which after simplification gives

$$\mathbb{E}\left(\|\widehat{\mathrm{cov}}(x,x)-\mathrm{cov}(x,x)\|_F^2\right) = \frac{1}{N}\sum_{m,m'}\left[\mathrm{var}(x_m x_{m'}) + 2\left[\mathbb{E}(x_m)\right]^2\mathrm{var}(x_{m'})\right]$$

$$+\frac{1}{N}\sum_{m,m'}\left[2\mathbb{E}(x_m)\mathbb{E}(x_{m'})\mathrm{cov}(x_m,x_{m'}) - 4\mathbb{E}(x_m)\mathrm{cov}(x_m x_{m'},x_{m'})\right] + O\left(\frac{1}{N^2}\right),$$

where in the last equality, by symmetry, the summation indexes $m$ and $m'$ can be exchanged. As $x_m \sim \mathrm{Poisson}(y_m)$, by the law of total expectation and law of total covariance, it follows, for $m \neq m'$ (and using the auxiliary expressions from Section D.4):

$$\mathrm{var}(x_m x_{m'}) = \mathbb{E}(x_m^2 x_{m'}^2) - [\mathbb{E}[x_m x_{m'}]]^2 = \mathbb{E}\left[\mathbb{E}(x_m^2 x_{m'}^2|y)\right] - \left[\mathbb{E}\left[\mathbb{E}(x_m x_{m'}|y)\right]\right]^2$$

$$= \mathbb{E}\left[y_m^2 y_{m'}^2 + y_m^2 y_{m'} + y_m y_{m'}^2 + y_m y_{m'}\right] - \left[\mathbb{E}(y_m y_{m'})\right]^2,$$

$$[\mathbb{E}(x_m)]^2\mathrm{var}(x_{m'}) = [\mathbb{E}(y_m)]^2\mathbb{E}(y_{m'}) + [\mathbb{E}(y_m)]^2\mathbb{E}(y_{m'}^2) - [\mathbb{E}(y_m)]^2[\mathbb{E}(y_{m'})]^2,$$

$$\mathbb{E}(x_m)\mathbb{E}(x_{m'})\mathrm{cov}(x_m,x_{m'}) = \mathbb{E}(y_m y_{m'})\mathbb{E}(y_m)\mathbb{E}(y_{m'}) - [\mathbb{E}(y_m)]^2[\mathbb{E}(y_{m'})]^2,$$

$$\mathbb{E}(x_m)\mathrm{cov}(x_m x_{m'},x_{m'}) = \mathbb{E}(y_m)\left[\mathbb{E}(y_m y_{m'}) + \mathbb{E}(y_m y_{m'}^2) - \mathbb{E}(y_m y_{m'})\mathbb{E}(y_{m'})\right].$$

Now, considering the $m = m'$ case, we have:

$$\mathrm{var}(x_m^2) = \mathbb{E}[\mathbb{E}(x_m^4|y)] - \left[\mathbb{E}[\mathbb{E}(x_m^2|y)]\right]^2$$

$$= \mathbb{E}\left[y_m^4 + 6y_m^3 + 7y_m^2 + y_m\right] - \left[\mathbb{E}\left[y_m^2 + y_m\right]\right]^2,$$

$$\mathbb{E}(x_m)\mathbb{E}(x_m)\mathrm{cov}(x_m,x_m) = \mathbb{E}(y_m)^2\left[\mathbb{E}(y_m^2) + \mathbb{E}(y_m) - [\mathbb{E}(y_m)]^2\right],$$

$$\mathbb{E}(x_m)\mathrm{cov}(x_m^2,x_m) = \mathbb{E}(y_m)\left[\mathbb{E}(y_m^3) + 3\mathbb{E}(y_m^2) + \mathbb{E}(y_m) - \mathbb{E}(y_m)\left[\mathbb{E}(y_m^2) + \mathbb{E}(y_m)\right]\right].$$

Substitution of $y_m = \sum_k D_{mk}\alpha_k$ gives the following

$$\mathbb{E}\left(\|\widehat{\mathrm{cov}}(x,x)-\mathrm{cov}(x,x)\|_F^2\right) = \frac{1}{N}\sum_{k,k',k'',k'''}\langle d_k, d_{k'}\rangle\langle d_{k''}, d_{k'''}\rangle\mathcal{A}_{kk'k''k'''}$$

$$+\frac{1}{N}\sum_{k,k',k''}\left[\langle d_k, d_{k'}\rangle\langle d_{k''}, \vec{1}\rangle\mathcal{B}_{kk'k''} + \langle d_k \circ d_{k'}, d_{k''}\rangle\mathcal{E}_{kk'k''}\right]$$

$$+\frac{1}{N}\sum_{k,k'}\left[\langle d_k, \vec{1}\rangle\langle d_{k'}, \vec{1}\rangle\mathbb{E}(\alpha_k\alpha_{k'}) + \langle d_k, d_{k'}\rangle\mathcal{F}_{kk'}\right]$$

$$+\sum_k\langle d_k, \vec{1}\rangle\mathbb{E}(\alpha_k) + O\left(\frac{1}{N^2}\right),$$

where $\vec{1}$ is the vector with all the elements equal to 1 and

$$\mathcal{A}_{kk'k''k'''} = \mathbb{E}(\alpha_k\alpha_{k'}\alpha_{k''}\alpha_{k'''}) - \mathbb{E}(\alpha_k\alpha_{k''})\mathbb{E}(\alpha_{k'}\alpha_{k'''}) + 2\mathbb{E}(\alpha_k)\mathbb{E}(\alpha_{k'})\mathbb{E}(\alpha_{k''}\alpha_{k'''})$$

$$- 2\mathbb{E}(\alpha_k)\mathbb{E}(\alpha_{k'})\mathbb{E}(\alpha_{k''})\mathbb{E}(\alpha_{k'''}) + 2\mathbb{E}(\alpha_k\alpha_{k''})\mathbb{E}(\alpha_{k'})\mathbb{E}(\alpha_{k'''}) - 2\mathbb{E}(\alpha_k)\mathbb{E}(\alpha_{k'})\mathbb{E}(\alpha_{k''})\mathbb{E}(\alpha_{k'''})$$

$$- 4\mathbb{E}(\alpha_k)\mathbb{E}(\alpha_{k'}\alpha_{k''}\alpha_{k'''}) + 4\mathbb{E}(\alpha_k)\mathbb{E}(\alpha_{k'}\alpha_{k''})\mathbb{E}(\alpha_{k'''}),$$

$$\mathcal{B}_{kk'k''} = 2\mathbb{E}(\alpha_k\alpha_{k'}\alpha_{k''}) + 2\mathbb{E}(\alpha_k)\mathbb{E}(\alpha_{k'})\mathbb{E}(\alpha_{k''}) - 4\mathbb{E}(\alpha_k)\mathbb{E}(\alpha_{k'}\alpha_{k''}),$$

$$\mathcal{E}_{kk'k''} = 4\mathbb{E}(\alpha_k\alpha_{k'}\alpha_{k''}) + 6\mathbb{E}(\alpha_k)\mathbb{E}(\alpha_{k'})\mathbb{E}(\alpha_{k''}) - 10\mathbb{E}(\alpha_k\alpha_{k'})\mathbb{E}(\alpha_{k''}),$$

$$\mathcal{F}_{kk'} = 6\mathbb{E}(\alpha_k\alpha_{k'}) - 5\mathbb{E}(\alpha_k)\mathbb{E}(\alpha_{k'}),$$

where we used the expressions from Section D.4.

## D.3 Expected squared error of the estimator $\widehat{S}$ for the GP/DICA cumulants

As the estimator $\widehat{S}$ (29) of $S$ (10) is unbiased, its expected squared error is

$$
\begin{aligned}
\mathbb{E}\left[\|\widehat{S} - S\|_F^2\right] =& \mathbb{E}\left[\left\|(\widehat{\mathrm{cov}}(x,x) - \mathrm{cov}(x,x)) + \left(\mathrm{diag}[\widehat{\mathbb{E}}(x)] - \mathrm{diag}\left[\mathbb{E}(x)\right]\right)\right\|_F^2\right] \\
=& \mathbb{E}\left[\|\widehat{\mathbb{E}}(x) - \mathbb{E}(x)\|_F^2\right] + \mathbb{E}\left[\|\widehat{\mathrm{cov}}(x,x) - \mathrm{cov}(x,x)\|_F^2\right] \\
&+ 2\sum_m \mathbb{E}\left[\left(\widehat{\mathbb{E}}(x_m) - \mathbb{E}(x_m)\right)\left(\widehat{\mathrm{cov}}(x_m,x_m) - \mathrm{cov}(x_m,x_m)\right)\right].
\end{aligned}
\tag{33}
$$

As $\widehat{\mathbb{E}}(x_m)$ and $\widehat{\mathrm{cov}}(x_m, x_m)$ are unbiased, the $m$-th element of the last sum is equal to

$$
\begin{aligned}
&\mathrm{cov}\left[\widehat{\mathbb{E}}(x_m), \widehat{\mathrm{cov}}(x_m, x_m)\right] \\
&= \frac{1}{N^2}\sum_{n,n'} \mathrm{cov}\left[x_{nm}, x_{n'm}^2\right] - \frac{1}{N^2(N-1)}\sum_{n,n',n''\neq n'} \mathrm{cov}\left[x_{nm}, x_{n'm}x_{n''m}\right] \\
&= \frac{1}{N^2}\sum_n \mathrm{cov}\left[x_{nm}, x_{nm}^2\right] - \frac{2}{N^2(N-1)}\sum_{n,n'\neq n} \mathrm{cov}\left[x_{nm}, x_{n'm}x_{nm}\right] + O\left(\frac{1}{N^2}\right) \\
&= \frac{1}{N}\mathbb{E}(x_m^3) - \frac{2}{N}\left(\mathbb{E}(x_m^2)\mathbb{E}(x_m) - [\mathbb{E}(x_m)]^3\right) + O\left(\frac{1}{N^2}\right) \\
&\leq \frac{1}{N}\mathbb{E}(x_m^3) + \frac{2}{N}[\mathbb{E}(x_m)]^3 + O\left(\frac{1}{N^2}\right) = \frac{1}{N}\left[\mathbb{E}(y_m^3) + 3\mathbb{E}(y_m^2) + \mathbb{E}(y_m) + 2[\mathbb{E}(y_m)]^3\right] + O\left(\frac{1}{N^2}\right),
\end{aligned}
$$

where we neglected the negative term $-\mathbb{E}(x_m^2)\mathbb{E}(x_m)$ for the inequality, and the last equality follows from the expressions in Section D.4. Further, the fact that $y_m = \sum_k D_{mk}\alpha_k$ gives

$$
\begin{aligned}
\sum_m \mathrm{cov}\left[\widehat{\mathbb{E}}(x_m), \widehat{\mathrm{cov}}(x_m, x_m)\right] =& \frac{1}{N}\sum_{k,k',k''}\langle d_k \circ d_{k'}, d_{k''}\rangle \mathcal{C}_{kk'k''} \\
&+ \frac{3}{N}\sum_{k,k'}\langle d_k, d_{k'}\rangle\mathbb{E}(\alpha_k\alpha_{k'}) + \frac{1}{N}\sum_k\langle d_k, \vec{1}\rangle\mathbb{E}(\alpha_k) + O\left(\frac{1}{N^2}\right),
\end{aligned}
$$

where $\circ$ denotes the element-wise Hadamard product and

$$
\mathcal{C}_{kk'k''} = \mathbb{E}(\alpha_k\alpha_{k'}\alpha_{k''}) + 2\mathbb{E}(\alpha_k)\mathbb{E}(\alpha_{k'})\mathbb{E}(\alpha_{k''}).
$$

Plugging this and the expressions for $\mathbb{E}(\|\widehat{\mathbb{E}}(x) - \mathbb{E}(x)\|_F^2)$ and $\mathbb{E}(\|\widehat{\mathrm{cov}}(x,x) - \mathrm{cov}(x,x)\|_F^2)$ from Sections D.1 and D.2, respectively, into (33) gives

$$
\mathbb{E}\left[\|\widehat{S} - S\|_F^2\right] = \frac{1}{N}\left[\sum_k\langle d_k, d_k\rangle\mathrm{var}(\alpha_k) + \sum_k\mathbb{E}(\alpha_k) + \sum_{k,k',k'',k'''}\langle d_k, d_{k'}\rangle\langle d_{k''}, d_{k'''}\rangle\mathcal{A}_{kk'k''k'''}\right] + O\left(\frac{1}{N^2}\right)
$$

$$
+ \frac{1}{N}\left[\sum_{k,k',k''}[\langle d_k, d_{k'}\rangle\mathcal{B}_{kk'k''} + 2\langle d_k \circ d_{k'}, d_{k''}\rangle\mathcal{C}_{kk'k''}] + \sum_{k,k'}(1 + 6\langle d_k, d_{k'}\rangle)\mathbb{E}(\alpha_k\alpha_{k'}) + 2\sum_k\mathbb{E}(\alpha_k)\right],
$$

where we used that, by the simplex constraint on the topics, $\langle d_k, \vec{1}\rangle = 1$ for all $k$. To analyze this expression in more details, let us now consider the GP model, i.e., $\alpha_k \sim \mathrm{Gamma}(c_k, b)$:

$$
\sum_{k,k',k'',k'''}\mathcal{A}_{kk'k''k'''} \leq \frac{30c_0^4 + 23c_0^3 + 14c_0^2 + 8c_0}{b^4}, \quad \text{and} \quad \sum_{k,k',k''}\mathcal{B}_{kk'k''} \leq \frac{6c_0^3 + 10c_0^2 + 4c_0}{b^3},
$$

$$
\sum_{k,k',k''}\mathcal{C}_{kk'k''} \leq \frac{7c_0^3 + 6c_0^2 + 2c_0}{b^3}, \quad \text{and} \quad \sum_{k,k',k''}\mathcal{E}_{kk'k''} \leq \frac{12c_0^3 + 10c_0^2 + 8c_0}{b^3},
$$

$$
\sum_{k,k'}\mathcal{F}_{kk'} \leq \frac{2c_0^2 + c_0}{b^2} \quad \text{and} \quad \sum_{k,k'}\mathbb{E}(\alpha_k\alpha_{k'}) \leq \frac{2c_0^2 + c_0}{b^2},
$$

where we used the expressions from Section D.4, which gives

$$\mathbb{E}\left[\|\widehat{S} - S\|_F^2\right] \leq \frac{\nu}{N}\left[\max_k \|d_k\|_2^2 \frac{c_0}{b^2} + \frac{c_0}{b} + \left(\max_{k,k'}\langle d_k, d_{k'}\rangle\right)^2 \max\left[\frac{c_0^4}{b^4}, \frac{c_0}{b^4}\right] + \max_{k,k'}\langle d_k, d_{k'}\rangle \max\left[\frac{c_0^3}{b^3}, \frac{c_0}{b^3}\right]\right]$$

$$+ \frac{\nu}{N}\left[\left(\max_{k,k',k''}\langle d_k \circ d_{k'}, d_{k''}\rangle\right)\max\left[\frac{c_0^3}{b^3}, \frac{c_0}{b^3}\right] + \left(1 + \max_{k,k'}\langle d_k, d_{k'}\rangle\right)\max\left[\frac{c_0^2}{b^2}, \frac{c_0}{b^2}\right]\right] + O\left(\frac{1}{N^2}\right),$$

where $\nu \leq 30$ is a universal constant. As, by the Cauchy-Schwarz inequality, $\max_{k,k'}\langle d_k, d_{k'}\rangle \leq \max_k \|d_k\|_2^2 =: \Delta_1$ and $\max_{k,k',k''}\langle d_k \circ d_{k'}, d_{k''}\rangle \leq \max_k \|d_k\|_\infty \|d_k\|_2^2 \leq \max_k \|d_k\|_2^3 =: \Delta_2$ (note that for the topics in the simplex, $\Delta_2 \leq \Delta_1$ as well as $\Delta_1^2 \leq \Delta_1$), it follows that

$$\mathbb{E}\left[\|\widehat{S} - S\|_F^2\right] \leq \frac{\nu}{N}\left[\Delta_1\left(\frac{L^2}{\bar{c}_0} + \frac{L^3}{\bar{c}_0^2}\right) + L + \Delta_1^2 \frac{L^4}{\bar{c}_0^3} + \frac{L^2}{\bar{c}_0^2} + \Delta_2 \frac{L^3}{\bar{c}_0^2}\right] + O\left(\frac{1}{N^2}\right)$$

$$\leq \frac{2\nu}{N}\frac{1}{\bar{c}_0^3}\left[\Delta_1^2 L^4 + \bar{c}_0 \Delta_1 L^3 + \bar{c}_0^2 L^2 + \bar{c}_0^3 L\right] + O\left(\frac{1}{N^2}\right),$$

where $\bar{c}_0 = \min(1, c_0) \leq 1$ and, from Section B.2, $c_0 = bL$ where $L$ is the expected document length. The second term $\bar{c}_0 \Delta_1 L^3$ cannot be dominant as the system $\bar{c}_0 \Delta_1 L^3 > \bar{c}_0^2 L^2$ and $\bar{c}_0 \Delta_1 L^3 > \Delta_1^2 L^4$ is infeasible. Also, with the reasonable assumption that $L \geq 1$, we also have that the 4th term $\bar{c}_0^3 L \leq \bar{c}_0^2 L^2$. Therefore,

$$\mathbb{E}\left[\|\widehat{S} - S\|_F^2\right] \leq \frac{3\nu}{N}\max\left[\Delta_1^2 L^4, \bar{c}_0^2 L^2\right] + O\left(\frac{1}{N^2}\right).$$

### D.4 Auxiliary expressions

As $\{x_m\}_{m=1}^M$ are conditionally independent given $y$ in the DICA model (3), we have the following expressions by using the law of total expectation for $m \neq m'$ and using the moments of the Poisson distribution with parameter $y_m$:

$$\mathbb{E}(x_m) = \mathbb{E}[\mathbb{E}(x_m|y_m)] = \mathbb{E}(y_m),$$
$$\mathbb{E}(x_m^2) = \mathbb{E}[\mathbb{E}(x_m^2|y_m)] = \mathbb{E}(y_m^2) + \mathbb{E}(y_m),$$
$$\mathbb{E}(x_m^3) = \mathbb{E}[\mathbb{E}(x_m^3|y_m)] = \mathbb{E}(y_m^3) + 3\mathbb{E}(y_m^2) + \mathbb{E}(y_m),$$
$$\mathbb{E}(x_m^4) = \mathbb{E}[\mathbb{E}(x_m^4|y_m)] = \mathbb{E}(y_m^4) + 6\mathbb{E}(y_m^3) + 7\mathbb{E}(y_m^2) + \mathbb{E}(y_m),$$
$$\mathbb{E}(x_m x_{m'}) = \mathbb{E}[\mathbb{E}(x_m x_{m'}|y)] = \mathbb{E}[\mathbb{E}(x_m|y_m)\mathbb{E}(x_{m'}|y_{m'})] = \mathbb{E}(y_m y_{m'}),$$
$$\mathbb{E}(x_m x_{m'}^2) = \mathbb{E}[\mathbb{E}(x_m x_{m'}^2|y)] = \mathbb{E}[\mathbb{E}(x_m|y_m)\mathbb{E}(x_{m'}^2|y_{m'})] = \mathbb{E}(y_m y_{m'}^2) + \mathbb{E}(y_m y_{m'}),$$
$$\mathbb{E}(x_m^2 x_{m'}^2) = \mathbb{E}[\mathbb{E}(x_m^2|y_m)\mathbb{E}(x_{m'}^2|y_{m'})] = \mathbb{E}(y_m^2 y_{m'}^2) + \mathbb{E}(y_m^2 y_{m'}) + \mathbb{E}(y_m y_{m'}^2) + \mathbb{E}(y_m y_{m'}).$$

Moreover, the moments of $\alpha_k \sim \text{Gamma}(c_k, b)$ are

$$\mathbb{E}(\alpha_k) = \frac{c_k}{b}, \quad \mathbb{E}(\alpha_k^2) = \frac{c_k^2 + c_k}{b^2}, \quad \mathbb{E}(\alpha_k^3) = \frac{c_k^3 + 3c_k^2 + 2c_k}{b^3}, \quad \mathbb{E}(\alpha_k^4) = \frac{c_k^4 + 6c_k^3 + 11c_k^2 + 6c_k}{b^4}, \quad \text{etc.}$$

### D.5 Analysis of whitening and recovery error

We can follow a similar analysis as in Appendix C of [15] to derive the topic recovery error given the sample estimate error. In particular, if we define the following sampling errors $E_S$ and $E_T$:

$$\|\widehat{S} - S\| \leq E_S,$$
$$\|\widehat{T}(u) - T(u)\| \leq \|u\|_2 E_T,$$

then the following form of their Lemma C.2 holds for both the LDA moments and the GP/DICA cumulants:

$$\|\widehat{W}\widehat{T}(\widehat{W}^\top u)\widehat{W}^\top - WT(W^\top u)W^\top\| \leq \nu\left[\frac{(\max_k \gamma_k)E_S}{\sigma_K(\widetilde{D})^2} + \frac{E_T}{\sigma_K(\widetilde{D})^3}\right], \tag{34}$$

where $\sigma_k(\cdot)$ denotes the $k$-th singular value of a matrix, $\nu$ is some universal constant, and in both cases $\widetilde{D}$ was defined such that $S = \widetilde{D}\widetilde{D}^\top$. For the LDA moments, $\gamma_k = 2\sqrt{\frac{c_0(c_0+1))}{c_k(c_0+2)^2}}$, whereas for the GP/DICA cumulants, $\gamma_k$ takes the simpler form $\gamma_k := \operatorname{cum}(\alpha_k)/[\operatorname{var}(\alpha_k)]^{3/2} = 2/\sqrt{c_k}$.

We note that the scaling for $S$ is $O(L^2)$ for the GP/DICA cumulants, in contrast to $O(1)$ for the LDA moments. Thus, to compare the upper bound (34) for the two types of moments, we need to put it in quantities which are common. In the first section of the Appendix C of [15], it was mentioned that $\sigma_K(\widetilde{D}) \geq \sqrt{\frac{c_{\min}}{c_0(c_0+1)}}\sigma_K(D)$ for the LDA moments, where $c_{\min} := \min_k c_k$. In contrast, for the GP/DICA cumulants, we can show that $\sigma_K(\widetilde{D}) \geq L\frac{\sqrt{c_{\min}}}{c_0}\sigma_K(D)$, where $L := c_0/b$ is the average length of a document in the GP model. Using this lower bound for the singular vector, we thus get the following bound in the case of the GP cumulant:

$$\|\widehat{W}\widehat{T}(\widehat{W}^\top u)\widehat{W}^\top - WT(W^\top u)W^\top\| \leq \frac{\nu}{c_{\min}^{3/2}}\left[\frac{E_S}{L^2}\frac{2c_0^2}{\left[\sigma_K(D)\right]^2} + \frac{E_T}{L^3}\frac{c_0^3}{\left[\sigma_K(D)\right]^3}\right]. \qquad (35)$$

The $c_{\min}^{3/2}$ factor is common for both the LDA moment and GP cumulant, but as we mentioned after Proposition 3.1, the sample error $E_S$ term gets divided by $L^2$ for the GP cumulant, as expected.

The recovery error bound in [15] is based on the bound (35), and thus by showing that the error $E_S/L^2$ for the GP cumulant is lower than the $E_S$ term for the LDA moment, we expect to also gain a similar gain for the recovery error, as the rest of the argument is the same for both types of moments (see Appendix C.2, C.3 and C.4 in [15] for the completion).

## E  Appendix. The LDA moments

### E.1  Our notation

The LDA moments were derived in [3]. Note that the full version of the paper with proofs appeared in [15] and a later version of the paper also appeared in [31]. In this section, we recall the form of the LDA moments using our notation. This section does not contain any novel results and is included for the reader's convenience. We also refer to this section when deriving the practical expressions for computation of the sample estimates of the LDA moments in Appendix F.4.

For deriving the LDA moments, a document is assumed to be composed of at least three tokens: $L \geq 3$. As the LDA generative model (1) is only defined *conditional* on the length $L$, this is not too problematic. But given that we present models in this paper which also model $L$, we mention for clarity that we can suppose that all expectations and probabilities defined below are implicitly conditioning on $L \geq 3$.[11] The theoretical LDA moments are derived only using the first three words $w_1$, $w_2$ and $w_3$ of a document. But note that since the words $w_\ell$'s are conditionally i.i.d. given $\theta$ (for $1 \leq \ell \leq L$), we have $M_3 := \mathbb{E}(w_1 \otimes w_2 \otimes w_3) = \mathbb{E}(w_{\ell_1} \otimes w_{\ell_2} \otimes w_{\ell_3})$ for any three distinct tokens $\ell_1$, $\ell_2$ and $\ell_3$. The tensor $M_3$ is thus symmetric, and could have been defined using any distinct $\ell_1$, $\ell_2$ and $\ell_3$ that are less than $L$. To highlight this arbitrary choice and to make the links with the U-statistics estimator presented later, we thus use generic distinct $\ell_1$, $\ell_2$ and $\ell_3$ in the definition of the LDA moments below, instead of $\ell_1 = 1$, $\ell_2 = 2$ and $\ell_3 = 1$ as in [3].

Using this notation, then by the law of total expectation and the properties of the Dirichlet distribution, the non-central moments[12] of the LDA model (1) take the form [3]:

$$M_1 = \mathbb{E}(w_{\ell_1}) = D\frac{c}{c_0}, \tag{36}$$

$$M_2 = \mathbb{E}(w_{\ell_1} w_{\ell_2}^\top) = \frac{c_0}{c_0 + 1} M_1 M_1^\top + \frac{1}{c_0(c_0 + 1)} D \operatorname{diag}(c) D^\top, \tag{37}$$

$$M_3 = \mathbb{E}(w_{\ell_1} \otimes w_{\ell_2} \otimes w_{\ell_3})$$

$$= \frac{c_0}{c_0 + 2} \left[ \mathbb{E}(w_{\ell_1} \otimes w_{\ell_2} \otimes M_1) + \mathbb{E}(w_{\ell_1} \otimes M_1 \otimes w_{\ell_3}) + \mathbb{E}(M_1 \otimes w_{\ell_2} \otimes w_{\ell_3}) \right],$$

$$- \frac{2c_0^3}{c_0(c_0 + 1)(c_0 + 2)} M_1 \otimes M_1 \otimes \widehat{M}_1 + \frac{2}{c_0(c_0 + 1)(c_0 + 2)} \sum_{k=1}^{K} c_k d_k \otimes d_k \otimes d_k. \tag{38}$$

where $\otimes$ denotes the tensor product.

Similarly to the GP/DICA cumulants (as discussed in Appendix C.3), moving the terms in the non-central moments (36), (37), (38), the following quantities are defined

$$(Pairs) = S := M_2 - \frac{c_0}{c_0 + 1} M_1 M_1^\top, \qquad \text{LDA S-moment} \tag{39}$$

$$(Triples) = T := M_3 - \frac{c_0}{c_0 + 2} \left[ \mathbb{E}(w_{\ell_1} \otimes w_{\ell_2} \otimes M_1) + \mathbb{E}(w_{\ell_1} \otimes M_1 \otimes w_{\ell_3}) + \mathbb{E}(M_1 \otimes w_{\ell_2} \otimes w_{\ell_3}) \right]$$

$$+ \frac{2c_0^2}{(c_0 + 1)(c_0 + 2)} M_1 \otimes M_1 \otimes M_1. \qquad \text{LDA T-moment} \tag{40}$$

Slightly abusing terminology, we refer to the entities $S$ and $T$ as the "LDA moments". They have the following diagonal structure

$$S = \frac{1}{c_0(c_0 + 1)} \sum_{k=1}^{K} c_k d_k d_k^\top, \tag{41}$$

$$T = \frac{2}{c_0(c_0 + 1)(c_0 + 2)} \sum_{k=1}^{K} c_k d_k \otimes d_k \otimes d_k. \tag{42}$$

Note however that this form of the LDA moments has a slightly different nature than the similar form (11) and (13) of the GP/DICA cumulants. Indeed, the former is the result of properties of the Dirichlet distribution, while the latter is the result of the independence of $\alpha$'s. However, one can think of the elements of a Dirichlet random vector as being almost independent (as, e.g., a Dirichlet random vector can be obtained from independent gamma variables through dividing each by their sum). Also, this closeness of the structures of the LDA moments and the GP cumulants can be explained by the closeness of the respective models as discussed in Section 2.

### E.2 Asymptotically unbiased finite sample estimators for the LDA moments

Given realizations $w_{n\ell}$, $n = 1, \ldots, N$, $\ell = 1, \ldots, L_n$, of the token random variable $w_\ell$, we now give the expressions for the finite sample estimates of $S$ (39) and $T$ (40) for the LDA model (and we re-write them as a function of the sample counts $x_n$).[13] We use the notation $\widehat{\mathbb{E}}$ below to express a U-statistics empirical expectation over the token within a documents, uniformly averaged over the whole corpus. For example, $\widehat{\mathbb{E}}(w_{\ell_1} \otimes w_{\ell_2} \otimes \widehat{M}_1) := \frac{1}{N} \sum_{n=1}^{N} \frac{1}{L_n(L_n-1)} \sum_{\ell_1=1}^{L_n} \sum_{\substack{\ell_2=1 \\ \ell_2 \neq \ell_1}}^{L_n} w_{\ell_1} \otimes w_{\ell_2} \otimes$

$\widehat{M}_1$.

$$\widehat{S} := \widehat{M}_2 - \frac{c_0}{c_0+1}\widehat{M}_1\widehat{M}_1^\top, \tag{43}$$

$$\widehat{T} := \widehat{M}_3 - \frac{c_0}{c_0+2}\left[\widehat{\mathbb{E}}(w_{\ell_1}\otimes w_{\ell_2}\otimes\widehat{M}_1) + \widehat{\mathbb{E}}(w_{\ell_1}\otimes\widehat{M}_1\otimes w_{\ell_3}) + \widehat{\mathbb{E}}(\widehat{M}_1\otimes w_{\ell_2}\otimes w_{\ell_3})\right]$$

$$+ \frac{2c_0^2}{(c_0+1)(c_0+2)}\widehat{M}_1\otimes\widehat{M}_1\otimes\widehat{M}_1, \tag{44}$$

where, as suggested in [4], unbiased U-statistics estimates of $M_1$, $M_2$ and $M_3$ are:

$$\widehat{M}_1 := \widehat{\mathbb{E}}(w_\ell) = \frac{1}{N}\sum_{n=1}^N\frac{1}{L_n}\sum_{\ell=1}^{L_n}w_{n\ell} = \frac{1}{N}\sum_{n=1}^N[\delta_1]_n x_n = \frac{1}{N}X\delta_1, \tag{45}$$

$$\widehat{M}_2 := \widehat{\mathbb{E}}(w_{\ell_1}w_{\ell_2}^\top) = \frac{1}{N}\sum_{n=1}^N\frac{1}{L_n(L_n-1)}\sum_{\ell_1=1}^{L_n}\sum_{\substack{\ell_2=1\\\ell_2\neq\ell_1}}^{L_n}w_{n\ell_1}w_{n\ell_2}^\top$$

$$= \frac{1}{N}\sum_{n=1}^N[\delta_2]_n\left(x_n x_n^\top - \sum_{\ell=1}^{L_n}w_{n\ell}w_{n\ell}^\top\right)$$

$$= \frac{1}{N}\sum_{n=1}^N[\delta_2]_n\left(x_n x_n^\top - \mathrm{diag}(x_n)\right)$$

$$= \frac{1}{N}\left[X\mathrm{diag}(\delta_2)X^\top - \mathrm{diag}(X\delta_2)\right], \tag{46}$$

$$\tag{47}$$

$$\widehat{M}_3 := \widehat{\mathbb{E}}(w_{\ell_1}\otimes w_{\ell_2}\otimes w_{\ell_3}) = \frac{1}{N}\sum_{n=1}^N\delta_{3n}\sum_{\ell_1=1}^{L_n}\sum_{\substack{\ell_2=1\\\ell_2\neq\ell_1}}^{L_n}\sum_{\substack{\ell_3=1\\\ell_3\neq\ell_2\\\ell_3\neq\ell_1}}^{L_n}w_{n\ell_1}\otimes w_{n\ell_2}\otimes w_{n\ell_3}$$

$$= \frac{1}{N}\sum_{n=1}^N[\delta_3]_n\left(x_n\otimes x_n\otimes x_n - \sum_{\ell=1}^{L_n}w_{n\ell}\otimes w_{n\ell}\otimes w_{n\ell}\right.$$

$$\left. - \sum_{\ell_1=1}^{L_n}\sum_{\substack{\ell_2=1\\\ell_2\neq\ell_1}}^{L_n}(w_{n\ell_1}\otimes w_{n\ell_1}\otimes w_{n\ell_2} + w_{n\ell_1}\otimes w_{n\ell_2}\otimes w_{n\ell_1} + w_{n\ell_1}\otimes w_{n\ell_2}\otimes w_{n\ell_2})\right)$$

$$= \frac{1}{N}\sum_{n=1}^N[\delta_3]_n\left(x_n\otimes x_n\otimes x_n + 2\sum_{m=1}^M x_{nm}(e_m\otimes e_m\otimes e_m)\right.$$

$$\left. - \sum_{m_1=1}^M\sum_{m_2=1}^M x_{nm_1}x_{nm_2}(e_{m_1}\otimes e_{m_1}\otimes e_{m_2} + e_{m_1}\otimes e_{m_2}\otimes e_{m_1} + e_{m_1}\otimes e_{m_2}\otimes e_{m_2})\right). \tag{48}$$

Here, the vectors $\delta_1$, $\delta_2$ and $\delta_3 \in \mathbb{R}^N$ are defined as $[\delta_1]_n := L_n^{-1}$; $[\delta_2]_n := (L_n(L_n-1))^{-1}$, i.e., $[\delta_2]_n = \left[\binom{L_n}{2}2!\right]^{-1}$ is the number of times to choose an ordered pair of tokens out of $L_n$ tokens; $[\delta_3]_n := (L_n(L_n-1)(L_n-2))^{-1}$, i.e., $[\delta_3]_n = \left[\binom{L_n}{3}3!\right]^{-1}$ is the number of times to choose an ordered triple of tokens out of $L_n$ tokens. Note that the vectors $\delta_1$, $\delta_2$, and $\delta_3$ have nothing to do with the Kronecker delta $\delta$.

For a vector $a \in \mathbb{R}^N$, we sometimes use notation $[a]_n$ to denote its $n$-th element. Similarly, for a matrix $A \in \mathbb{R}^{M\times N}$ we use notation $[A]_{mn}$ to denote its $(m,n)$-th element.

There is a slight abuse of notation in the expressions above as $w_\ell$ is sometimes treated as a random variable (i.e., in $\widehat{\mathbb{E}}(w_\ell)$, $\widehat{\mathbb{E}}(w_{\ell_1} w_{\ell_2}^\top)$, etc.) and sometimes as its realization. However, the difference is clear from the context.

## F   Appendix. Practical aspects and implementation details

### F.1   Whitening of S and dimensionality reduction

The algorithms from Section 4 require the computation of a whitening matrix $W$ of $S$. Due to the similar diagonal structure ((41) and (11)) of the matrix S for both the LDA moments (39) and the GP/DICA cumulants (10), the computation of a whitening matrix is exactly the same in both cases.

By a whitening matrix, we mean a matrix $W \in \mathbb{R}^{K \times M}$ (in practice, $M \gg K$) that does not only whiten $S \in \mathbb{R}^{M \times M}$, but also reduces its dimensionality such that[14] $WSW^\top = I_K$.

Let $S = U \Sigma U^\top$ be an orthogonal eigendecomposition of the symmetric matrix $S$. Let $\Sigma_{1:K}$ denotes the diagonal matrix that contains the largest $K$ eigenvalues[15] of $S$ on its diagonal and let $U_{1:K}$ be a matrix with the respective eigenvalues in its columns. Then, a whitening matrix is

$$W = \Sigma_{1:K}^{\dagger 1/2} U_{1:K}^\top, \tag{49}$$

where $\Sigma_{1:K}^{\dagger 1/2}$ is a diagonal matrix constructed from $\Sigma_{1:K}$ by taking the inverse and the square root of its non-zero diagonal values ($\dagger$ stands for the pseudo-inverse).

In practice, when only a finite sample estimator $\widehat{S}$ of $S$ is available, the following finite sample estimator $\widehat{W}$ of $W$ can be introduced

$$\widehat{W} := \widehat{\Sigma}_{1:K}^{\dagger 1/2} \widehat{U}_{1:K}^\top, \tag{50}$$

where $\widehat{S} = \widehat{U} \widehat{\Sigma} \widehat{U}^\top$.

### F.2   Computation of the finite sample estimators of the GP/DICA cumulants

In this section, we present efficient formulas for computation of the finite sample estimate (see Appendix C.4 for the definition of $\widehat{T}$) of $\widehat{W}\widehat{T}(v)\widehat{W}^\top$ for the GP/DICA models. The construction of the finite sample estimator $\widehat{W}$ is discussed in Appendix F.1, while the computation of $\widehat{S}$ (29) is straightforward.

By plugging the definition of the tensor $\widehat{T}$ (30) in the formula (16) for the projection of a tensor onto a vector, we obtain for a given $v \in \mathbb{R}^M$:

$$
\begin{aligned}
\left[\widehat{T}(v)\right]_{m_1 m_2} &= \sum_{m_3} \widehat{\mathrm{cum}}(x_{m_1}, x_{m_2}, x_{m_3}) v_{m_3} + 2 \sum_{m_3} \delta(m_1, m_2, m_3) \widehat{\mathbb{E}}(x_{m_3}) v_{m_3} \\
&\quad - \sum_{m_3} \delta(m_2, m_3) \widehat{\mathrm{cov}}(x_{m_1}, x_{m_2}) v_{m_3} \\
&\quad - \sum_{m_3} \delta(m_1, m_3) \widehat{\mathrm{cov}}(x_{m_1}, x_{m_2}) v_{m_3} \\
&\quad - \sum_{m_3} \delta(m_1, m_2) \widehat{\mathrm{cov}}(x_{m_1}, x_{m_3}) v_{m_3} \\
&= \sum_{m_3} \widehat{\mathrm{cum}}(x_{m_1}, x_{m_2}, x_{m_3}) v_{m_3} + 2 \delta(m_1, m_2) \widehat{\mathbb{E}}(x_{m_1}) v_{m_1} \\
&\quad - \widehat{\mathrm{cov}}(x_{m_1}, x_{m_2}) v_{m_2} - \widehat{\mathrm{cov}}(x_{m_1}, x_{m_2}) v_{m_1} - \delta(m_1, m_2) \sum_{m_3} \widehat{\mathrm{cov}}(x_{m_1}, x_{m_3}) v_{m_3}.
\end{aligned}
$$

This gives the following for the expression $\widehat{W} \widehat{T}(v) \widehat{W}^\top$:

$$
\begin{aligned}
\left[\widehat{W} \widehat{T}(v) \widehat{W}^\top\right]_{k_1 k_2} &= \widehat{W}_{k_1}^\top \widehat{T}(v) \widehat{W}_{k_2} \\
&= \sum_{m_1, m_2, m_3} \widehat{\mathrm{cum}}(x_{m_1}, x_{m_2}, x_{m_3}) v_{m_3} \widehat{W}_{k_1 m_1} \widehat{W}_{k_2 m_2} \\
&\quad + 2 \sum_{m_1, m_2} \delta(m_1, m_2) \widehat{\mathbb{E}}(x_{m_1}) v_{m_1} \widehat{W}_{k_1 m_1} \widehat{W}_{k_2 m_2} \\
&\quad - \sum_{m_1, m_2} \widehat{\mathrm{cov}}(x_{m_1}, x_{m_2}) v_{m_2} \widehat{W}_{k_1 m_1} \widehat{W}_{k_2 m_2} \\
&\quad - \sum_{m_1, m_2} \widehat{\mathrm{cov}}(x_{m_1}, x_{m_2}) v_{m_1} \widehat{W}_{k_1 m_1} \widehat{W}_{k_2 m_2} \\
&\quad - \sum_{m_1, m_3} \widehat{\mathrm{cov}}(x_{m_1}, x_{m_3}) v_{m_3} \widehat{W}_{k_1 m_1} \widehat{W}_{k_2 m_1}.
\end{aligned}
$$

where $\widehat{W}_k$ denotes the $k$-th row of $\widehat{W}$ as a column vector. By further plugging in the expressions (31) for the unbiased finite sample estimates of $\widehat{\mathrm{cov}}$ and $\widehat{\mathrm{cum}}$, we further get

$$
\begin{aligned}
\left[\widehat{W} \widehat{T}(v) \widehat{W}^\top\right]_{k_1 k_2} &= \frac{N}{(N-1)(N-2)} \sum_n \left\langle \widehat{W}_{k_1}, x_n - \widehat{\mathbb{E}}(x) \right\rangle \left\langle \widehat{W}_{k_2}, x_n - \widehat{\mathbb{E}}(x) \right\rangle \left\langle v, x_n - \widehat{\mathbb{E}}(x) \right\rangle \\
&\quad + 2 \sum_m \widehat{\mathbb{E}}(x_m) v_m \widehat{W}_{k_1 m} \widehat{W}_{k_2 m} \\
&\quad - \frac{1}{N-1} \sum_n \left\langle \widehat{W}_{k_1}, x_n - \widehat{\mathbb{E}}(x) \right\rangle \left\langle v \circ \widehat{W}_{k_2}, x_n - \widehat{\mathbb{E}}(x) \right\rangle \\
&\quad - \frac{1}{N-1} \sum_n \left\langle v \circ \widehat{W}_{k_1}, x_n - \widehat{\mathbb{E}}(x) \right\rangle \left\langle \widehat{W}_{k_2}, x_n - \widehat{\mathbb{E}}(x) \right\rangle \\
&\quad - \frac{1}{N-1} \sum_n \left\langle \widehat{W}_{k_1} \circ \widehat{W}_{k_2}, x_n - \widehat{\mathbb{E}}(x) \right\rangle \left\langle v, x_n - \widehat{\mathbb{E}}(x) \right\rangle,
\end{aligned}
$$

where $\circ$ denotes the element-wise Hadamard product. Introducing the counts matrix $X \in \mathbb{R}^{M \times N}$ where each element $X_{mn}$ is the count of the $m$-th word in the $n$-th document (note, the matrix $X$

contain the vector $x_n$ in the $n$-th column), we further simplify the above expression

$$\widehat{W}\widehat{T}(v)\widehat{W}^\top = \frac{N}{(N-1)(N-2)}(\widehat{W}X)\mathrm{diag}[X^\top v](\widehat{W}X)^\top$$
$$+ \frac{N}{(N-1)(N-2)}\left\langle v, \widehat{\mathbb{E}}(x)\right\rangle\left[2N(\widehat{W}\widehat{\mathbb{E}}(x))(\widehat{W}\widehat{\mathbb{E}}(x))^\top - (\widehat{W}X)(\widehat{W}X)^\top\right]$$
$$- \frac{N}{(N-1)(N-2)}\left[\widehat{W}X(X^\top v)(\widehat{W}\widehat{\mathbb{E}}(x))^\top + \widehat{W}\widehat{\mathbb{E}}(x)(\widehat{W}X(X^\top v))^\top\right]$$
$$+ 2\widehat{W}\mathrm{diag}[v \circ \widehat{\mathbb{E}}(x)]\widehat{W}^\top$$
$$- \frac{1}{N-1}\left[(\widehat{W}X)(\widehat{W}\mathrm{diag}(v)X)^\top + (\widehat{W}\mathrm{diag}(v)X)(\widehat{W}X)^\top + \widehat{W}\mathrm{diag}[X(X^\top v)]\widehat{W}^\top\right]$$
$$+ \frac{N}{N-1}\left[(\widehat{W}\widehat{\mathbb{E}}(x))(\widehat{W}\mathrm{diag}[v]\widehat{\mathbb{E}}(x))^\top + (\widehat{W}\mathrm{diag}[v]\widehat{\mathbb{E}}(x))(\widehat{W}\widehat{\mathbb{E}}(x))^\top\right]$$
$$+ \frac{N}{N-1}\left\langle v, \widehat{\mathbb{E}}(x)\right\rangle\widehat{W}\mathrm{diag}[\widehat{\mathbb{E}}(x)]\widehat{W}^\top. \tag{51}$$

A more compact way to write down expression (51) is as follows

$$\widehat{W}\widehat{T}(v)\widehat{W}^\top = \frac{N}{(N-1)(N-2)}\left[T_1 + \langle v, \widehat{\mathbb{E}}(x)\rangle(T_2 - T_3) - (T_4 + T_4^\top)\right]$$
$$+ \frac{1}{N-1}\left[T_5 + T_5^\top - T_6 - T_6^\top + \widehat{W}\mathrm{diag}(a)\widehat{W}^\top\right], \tag{52}$$

where

$$T_1 = (\widehat{W}X)\mathrm{diag}[X^\top v](\widehat{W}X)^\top,$$
$$T_2 = 2N(\widehat{W}\widehat{\mathbb{E}}(x))(\widehat{W}\widehat{\mathbb{E}}(x))^\top,$$
$$T_3 = (\widehat{W}X)(\widehat{W}X)^\top,$$
$$T_4 = \widehat{W}X(X^\top v)(\widehat{W}\widehat{\mathbb{E}}(x))^\top,$$
$$T_5 = (\widehat{W}X)(\widehat{W}\mathrm{diag}(v)X)^\top,$$
$$T_6 = (\widehat{W}\mathrm{diag}(v)\widehat{\mathbb{E}}(x))(\widehat{W}\widehat{\mathbb{E}}(x))^\top,$$
$$a = 2(N-1)[v \circ \widehat{\mathbb{E}}(x)] + \langle v, \widehat{\mathbb{E}}(x)\rangle\widehat{\mathbb{E}}(x) - X(X^\top v).$$

### F.3 Computational complexity of the GP/DICA T-cumulant estimator (52)

When computing the T-cumulant $P$ times with the formula above, the following terms are dominant: $O(RNK) + O(NK^2) + O(MK)$, where $R$ is the largest number of unique words (non-zero counts) in a document over the corpus. In practice, almost always $K < M < N$, which gives the overall complexity of $P$ computations of the estimator (52) to be equal to $O(PRNK) + O(PNK^2)$.

### F.4 Computation of the finite sample estimators of the LDA moments

In this section, we present efficient formulas for computation of the finite sample estimate (see Appendix E.2 for the definition of $\widehat{T}$) of $\widehat{W}\widehat{T}(v)\widehat{W}^\top$ for the LDA model. Note that the construction of the sample estimator $\widehat{W}$ of a whitening matrix $W$ is discussed in Appendix F.1). The computation of $\widehat{S}$ (43) is straightforward. This approach to efficient implementation was discussed in [4], however, to the best of our knowledge, the final expressions were not explicitly stated before. All derivations are straightforward, but quite tedious.

By analogy with the GP/DICA case, a projection (16) of the tensor $\widehat{T} \in \mathbb{R}^{M \times M \times M}$ (44) onto some vector $v \in \mathbb{R}^M$ in the LDA is

$$\left[\widehat{T}(v)\right]_{m_1 m_2} = \sum_{m_3=1}^{M} \left[\widehat{M}_3\right]_{m_1 m_2 m_3} v_{m_3} + \frac{2c_0^2}{(c_0+1)(c_0+2)} \sum_{m_3} [\widehat{M}_1]_{m_1}[\widehat{M}_1]_{m_2}[\widehat{M}_1]_{m_3} v_{m_3}$$

$$- \frac{c_0}{c_0+2} \sum_{m_3=1}^{M} \left[\widehat{\mathbb{E}}(w_{\ell_1} \otimes w_{\ell_2} \otimes \widehat{M}_1) + \widehat{\mathbb{E}}(w_{\ell_1} \otimes \widehat{M}_1 \otimes w_{\ell_3}) + \widehat{\mathbb{E}}(\widehat{M}_1 \otimes w_{\ell_2} \otimes w_{\ell_3})\right]_{m_1 m_2 m_3} v_{m_3}.$$

Plugging in the expression (48) for an unbiased sample estimate $\widehat{M}_3$ of $M_3$, we get

$$\left[\widehat{T}(v)\right]_{m_1 m_2} = \frac{1}{N}\sum_{n=1}^{N}[\delta_3]_n \left( x_{nm_1} x_{nm_2} \langle x_n, v\rangle + 2\sum_{m_3} \delta(m_1, m_2, m_3) x_{nm_3} v_{m_3} \right)$$

$$- \frac{1}{N}\sum_{n=1}^{N}[\delta_3]_n \sum_{m_3=1}^{M}\left[\sum_{i,j=1}^{M} x_{ni} x_{nj}\left(e_i \otimes e_i \otimes e_j + e_i \otimes e_j \otimes e_i + e_i \otimes e_j \otimes e_j\right)\right]_{m_1 m_2 m_3} v_{m_3}$$

$$+ \frac{2c_0^2}{(c_0+1)(c_0+2)}[\widehat{M}_1]_{m_1}[\widehat{M}_1]_{m_2}\left\langle \widehat{M}_1, v\right\rangle$$

$$- \frac{c_0}{c_0+2}\left([\widehat{M}_2]_{m_1 m_2}\left\langle\widehat{M}_1, v\right\rangle + \sum_{m_3=1}^{M}\left([\widehat{M}_2]_{m_1 m_3}[\widehat{M}_1]_{m_2} v_{m_3} + [\widehat{M}_2]_{m_2 m_3}[\widehat{M}_1]_{m_1} v_{m_3}\right)\right),$$

where $e_1, e_2, \ldots, e_M$ denote the canonical vectors of $\mathbb{R}^M$ (i.e., the columns of the identity matrix $I_M$). Further, this gives the following for the expression $\widehat{W}\widehat{T}(v)\widehat{W}^\top$:

$$\left[\widehat{W}\widehat{T}(v)\widehat{W}^\top\right]_{k_1 k_2} = \frac{1}{N}\sum_{n=1}^{N}[\delta_3]_n \left( \langle x_n, v\rangle \left\langle x_n, \widehat{W}_{k_1}\right\rangle \left\langle x_n, \widehat{W}_{k_2}\right\rangle + 2\sum_{m=1}^{M} x_{nm} v_m \widehat{W}_{k_1 m}\widehat{W}_{k_2 m} \right)$$

$$- \frac{1}{N}\sum_{n=1}^{N}\delta_{3n}\sum_{i,j=1}^{M} x_{ni} x_{nj}\left(\widehat{W}_{k_1 i}\widehat{W}_{k_2 i} v_j + \widehat{W}_{k_1 i}\widehat{W}_{k_2 j} v_i + \widehat{W}_{k_1 i}\widehat{W}_{k_2 j} v_j\right)$$

$$- \frac{c_0}{c_0+2}\left(\left\langle\widehat{W}_{k_1}, \left[\widehat{M}_2\right]\widehat{W}_{k_2}\right\rangle + \left\langle\widehat{W}_{k_1}, \widehat{M}_2 v\right\rangle\left\langle\widehat{M}_1 \widehat{W}_{k_2}\right\rangle + \left\langle\widehat{W}_{k_2}, \widehat{M}_2 v\right\rangle\left\langle\widehat{M}_1, \widehat{W}_{k_1}\right\rangle\right)$$

$$+ \frac{2c_0^2}{(c_0+1)(c_0+2)}\left\langle\widehat{M}_1, \widehat{W}_{k_1}\right\rangle\left\langle\widehat{M}_1, \widehat{W}_{k_2}\right\rangle\left\langle\widehat{M}_1, v\right\rangle,$$

where $\widehat{W}_k$ denotes the $k$-th row of $\widehat{W}$ as a column-vector. This further simplifies to

$$\widehat{W}\widehat{T}(v)\widehat{W}^\top = \frac{1}{N}(\widehat{W}X)\text{diag}\left[(X^\top v)\circ\delta_3\right](\widehat{W}X)^\top$$

$$+ \frac{1}{N}\widehat{W}\text{diag}\left[2[(X\delta_3)\circ v] - X[(X^\top v)\circ\delta_3]\right]\widehat{W}^\top$$

$$- \frac{1}{N}(\widehat{W}\text{diag}[v]X)\text{diag}[\delta_3](\widehat{W}X)^\top$$

$$- \frac{1}{N}(\widehat{W}X)\text{diag}[\delta_3](\widehat{W}\text{diag}[v]X)^\top$$

$$- \frac{c_0}{c_0+2}\left[\left\langle\widehat{M}_1, v\right\rangle(\widehat{W}\widehat{M}_2\widehat{W}^\top) + (\widehat{W}(\widehat{M}_2 v))(\widehat{W}\widehat{M}_1)^\top + (\widehat{W}\widehat{M}_1)(\widehat{W}(\widehat{M}_2 v))^\top\right]$$

$$+ \frac{2c_0^2}{(c_0+1)(c_0+2)}\left\langle\widehat{M}_1, v\right\rangle(\widehat{W}\widehat{M}_1)(\widehat{W}\widehat{M}_1)^\top. \tag{53}$$

A more compact representation gives:

$$\widehat{W}\widehat{T}(v)\widehat{W}^\top = \frac{1}{N}\left[T_1 + T_2 - T_3 - T_3^\top\right] - \frac{c_0}{c_0+2}\left[\langle\widehat{M}_1, v\rangle(\widehat{W}\widehat{M}_2\widehat{W}^\top) + T_4 + T_4^\top\right]$$

$$+ \frac{2c_0^2}{(c_0+1)(c_0+2)}\langle\widehat{M}_1, v\rangle(\widehat{W}\widehat{M}_1)(\widehat{W}\widehat{M}_1)^\top, \tag{54}$$

where

$$T_1 = (\widehat{W}X)\mathrm{diag}\left[(X^\top v) \circ \delta_3\right](\widehat{W}X)^\top,$$
$$T_2 = \widehat{W}\mathrm{diag}\left[2[(X\delta_3) \circ v] - X[(X^\top v) \circ \delta_3]\right]\widehat{W}^\top,$$
$$T_3 = [\widehat{W}\mathrm{diag}(v)X]\mathrm{diag}(\delta_3)(\widehat{W}X)^\top,$$
$$T_4 = [\widehat{W}(\widehat{M}_2 v)](\widehat{W}\widehat{M}_1)^\top.$$

### F.5 Computational complexity of the LDA T-moment estimator (54)

By analogy with Appendix F.3, the computational complexity of the T-moment is $O(RNK) + O(NK^2)$. However, in practice we noticed that the computation of (52) is slightly faster for larger datasets than the computation of (54) (although the code for both was equally well optimized). This means that the constants in $O(RNK) + O(NK^2)$ for the LDA T-moment are, probably, slightly larger than for the GP/DICA T-cumulant.

### F.6 Estimation of the model parameters for GP/DICA model

Below we briefly discuss the recovery of the model parameters for the GP/DICA and LDA models from a joint diagonalization matrix $A \in \mathbb{R}^{K \times M}$ estimated in Algorithm 1. This matrix has the property that $AD$ should be approximately diagonal up to a permutation of the columns of $D$. The standard approach [3] of taking the pseudo-inverse of $A$ to get an estimate of the topic matrix $D$ has a problem that it does not preserve the simplex constraint of the topics (in particular, the non-negativity of $\widetilde{D}$). Due to the space constraints, we do not discuss this issue here, but we observed experimentally that this can potentially significantly deteriorate performance of all moment matching algorithms for topic models considered in this paper. We made an attempt to solve this problem by integrating the non-negativity constraint into the Jacobi-updates procedure of the orthogonal joint diagonalization algorithm, but the obtained results did not lead to any significant improvement. Therefore, in our experiments for both GP/DICA cumulants and LDA moments, we estimate the topic matrix by thresholding the negative values of the pseudo-inverse of $A$:

$$\widehat{d}_k := \tau_k \max(0, [A^\dagger]_{:k})/\|\max(0, [A^\dagger]_{:k})\|_1,$$

where $[A^\dagger]_{:k}$ is the $k$-th column of the pseudo-inverse $A^\dagger$ of $A$, and $\tau_k = \pm 1$ set to $-1$ if $[A^\dagger]_{:k}$ has more negative than positive values. This might not be the best option, and we leave this issue for the future research.

To estimate the parameters for the prior distribution over topic intensities $\alpha_k$ for the DICA model (4), we use the diagonalized form of the projected tensor from (17) and relate it to the output diagonal elements $a_p$ for the $p$-th projection:

$$[a_p]_k = \widetilde{t}_k\langle z_k, u_p\rangle = \frac{t_k}{s_k^{3/2}}\langle z_k, u_p\rangle = \frac{\mathrm{cum}(\alpha_k, \alpha_k, \alpha_k)}{[\mathrm{var}(\alpha_k)]^{3/2}}\left\langle \tau_k\widetilde{d}_k, W^\top u_p\right\rangle, \qquad (55)$$

where $\widetilde{d}_k = \tau_k\max(0, [A^\dagger]_{:k})$. This formula is valid for any prior on $\alpha_k$ in the DICA model. For the GP model (3) where $\alpha_k \sim \mathrm{Gamma}(c_k, b)$, we have that $\mathrm{var}(\alpha_k) = \frac{c_k}{b^2}$ and $\mathrm{cum}(\alpha_k, \alpha_k, \alpha_k) = \frac{2c_k}{b^3}$, and thus $\widetilde{t}_k = \frac{2}{\sqrt{c_k}}$, which enables us to estimate $c_k$. Plugging this value of $\widetilde{t}_k$ in (55), and solving for $c_k$ gives the following expression:

$$c_k = \frac{4\left\langle \widetilde{d}_k, W^\top u_p\right\rangle^2}{[a_p]_k^2}.$$

By replacing the quantities on the RHS with their estimated ones, we get one estimate for $c_k$ per projection. We use as our final estimate the average estimate over the projections:

$$\widehat{c}_k := \frac{1}{P}\sum_{p=1}^{P}\frac{4\left\langle \widetilde{d}_k, \widehat{W}^\top u_p\right\rangle^2}{[a_p]_k^2}. \qquad (56)$$

Reusing the properties of the length of documents for the GP model as described in Appendix B.2, we finally use the following estimates for rate parameter $b$ of the gamma distribution:

$$\widehat{b} := \frac{\widehat{c}_0}{\widehat{L}}, \tag{57}$$

where $\widehat{c}_0 := \sum_k \widehat{c}_k$ and $\widehat{L}$ is the average document length in the corpus.

By analogy, similar formulas for the estimation of the Dirichlet parameter $c$ of the LDA model can be derived and are a straightforward extension of the expression in [3].

## G  Appendix. Complexity of algorithms and details on the experiments

### G.1  Code and complexity

Our (mostly Matlab) implementations of the diagonalization algorithms (JD, Spec, and TPM) for both the GP/DICA cumulants and LDA moments are available online.[16] Moreover, all datasets and code for reproducing our experiments are available.[17] To our knowledge, no efficient implementation of these algorithms was available for LDA. Each experiment was run in a single thread.

The bottleneck for the spectral, JD, and TPM algorithms is the computation of the cumulants/moments. However, the expressions (52) and (54) provide efficient formulas for fast computation of the GP/DICA cumulants and LDA moments ($O(RNK + NK^2)$, where $R$ is the largest number of non-zeros in the count vector $x$ over all documents, see Appendix F.3 and F.5), which makes even the Matlab implementation fast for large datasets. Since all diagonalization algorithms (spectral, JD, TPM) perform the whitening step once, it is sufficient to compare their complexities by the number of times the cumulants/moments are computed.

**Spectral.** The spectral algorithm estimates the cumulants/moments only once leading to $O(NK(R + K))$ complexity and, therefore, is the fastest.

**JD.** For JD, rather than estimating $P$ cumulants/moments separately, one can jointly estimate these values by precomputing and reusing some terms (e.g., $WX$). However, the complexity is still $O(PNK(R + K))$, although in practice it is sufficient to have $P = K$ or even smaller.

**TPM.** For TPM some parts of the cumulants/moments can also be precomputed, but as TPM normally does many more iterations than $P$, it can be significantly slower. In general, the complexity of TPM can be significantly influenced by the parameter's setting. There are two main parameters: $L_{tpm}$ is the number of random restarts within one deflation step and $N_{tpm}$ is the maximum number of iterations for each of $L_{tpm}$ random restarts (different from $N$ and $L$). Some restarts converge very fast (in much less than $N_{tpm}$ iterations), while others are slow. Moreover, as follows from theoretical results [4] and, as we observed in practice, the restarts which converge to a good solution converge fast, while slow restarts, normally, converge to a worse solution. Nevertheless, in the worst case, the complexity is $O(N_{tpm}L_{tpm}NK(R + K))$.

Note that for the experiment on Figure 1, $L_{tpm} = 10$ and $N_{tpm} = 100$ and the run with the best objective is chosen. We believe that these values are reasonable in a sense that they provide good accuracy solution ($\varepsilon = 10^{-5}$ for the norm of the difference of the vectors from the previous and the current iteration) in a little number of iterations, however, they may not be the best ones.

**JD implementation.** For the orthogonal joint diagonalization algorithm, we implemented a faster C++ version of the previous Matlab implementation (http://perso.telecom-paristech. fr/~cardoso/Algo/Joint_Diag/joint_diag_r.m) by J.-F. Cardoso. Moreover, the orthogonal joint diagonalization routine can be initialized in different ways: (a) with the $K \times K$ identity matrix or (b) with a random orthogonal $K \times K$ matrix. We tried different options and in nearly all cases the algorithm converged to the same solution, implying that initialization with the identity matrix is sufficient.

**Whitening matrix.** For the large vocabulary size $M$, the computation of a whitening matrix can be expensive (in terms of both memory and time). One possible solution would be to reduce the

|         | min   | mean | max   |
|---------|-------|------|-------|
| JD-GP   | 148   | 192  | 247   |
| JD-LDA  | 252   | 284  | 366   |
| JD(k)-GP | 157  | 190  | 247   |
| JD(k)-LDA | 264 | 290  | 318   |
| JD(f)-GP | 1628 | 1846 | 2058  |
| JD(f)-LDA | 2545 | 2649 | 2806 |
| Spec-GP | 101   | 107  | 111   |
| Spec-LDA | 107  | 140  | 193   |
| TPM-GP  | 1734  | 2393 | 2726  |
| TPM-LDA | 12723 | 1646 | 19356 |

Table 1: The running times in seconds of the algorithms from Figure 1, corresponds to the case when $N = 50,000$. Each algorithm was run 5 times, so the times in the table display the minimum (min), mean, and maximum (max) time.

vocabulary size with, e.g., TF-IDF score, which is a standard practice in the topic modeling context. Another option is using a stochastic eigendecomposition (see, e.g., [33]) to approximate the whitening matrix.

**Variational inference.** For variational inference, we used the code of D. Blei and modified it for the estimation of a non-symmetric Dirichlet prior $c$, which is known to be important [35]. The default values of the tolerance/maximum number of iterations parameters are used for variational inference. The computational complexity of one iteration for one document of the variational inference algorithm is $O(RK)$, where $R$ is the number of non-zeros in the count vector for this document, which is then performed a significant number of times for each document.

## G.2 Runtimes of the algorithms

In Table 1, we present the running times of the algorithms from Section 5.1. JD and JD(k) are significantly faster than JD(f) as expected, although the performance in terms of the $\ell_1$-error is nearly the same for all of them. This indicates that preference should be given to the JD or JD(k) algorithms.

The running time of all LDA-algorithms is higher than the one of the GP/DICA-algorithms. This indicates that the computational complexity of the LDA-moments is slightly higher than the one of the GP/DICA-cumulants (compare, e.g., the times for the spectral algorithm which almost completely consist of the computation of the moments/cumulants). Moreover, the runtime of TPM-LDA is significantly higher (half an hour vs. several hours) than the one of TPM-GP/DICA. This can be explained by the fact that the LDA-moments have more noise than the GP/DICA-cumulants and, hence, the algorithm is slower. Interestingly, all versions of JD algorithm are not that sensitive to noise.

Computation of a whitening matrix is roughly 30 sec (this time is the same for all algorithms and is included in the numbers above).

## G.3 Initialization of the parameter $c_0$ for the LDA moments

The construction of the LDA moments requires the parameter $c_0$, which is not trivial to set in the unsupervised setting of topic modeling, especially taking into account the complexity of evaluation for topic models [16]. For the semi-synthetic experiments, the true value of $c_0$ is provided to the algorithms. It means that the LDA moments, in this case, have access to some oracle information, which in practice is never available. For real data experiments, $c_0$ is set to the value obtained with variational inference. Experiments in Appendix G.4 show that this choice was somewhat important, however, this requires more thorough investigation.

## G.4 The LDA moments vs parameter $c_0$

In this section, we experimentally investigate dependence of the LDA moments on the parameter $c_0$. In Figure 5, the joint diagonalization algorithm with the LDA moment is compared for different values of $c_0$ provided to the algorithm. The data is generated similarly to Figure 2. The experiment

Figure 5: Performance of the LDA moments depending on the parameter $c_0$. $D$ and $c$ are learned from the AP dataset for $K = 10$ and $K = 50$ and true $c_0 = 1$. JD-GP(10) for $K = 10$ and JD-GP(50) for $K = 50$. Number of sampled documents $N = 20,000$. For the error bars, each dataset is resampled 5 times. Data **(left)**: *GP* sampling; **(right)**: *LDAfix(200)* sampling. *Note*: a smaller value of the $\ell_1$-error is better.

Figure 6: Comparison of the $\ell_1$- and $\ell_2$- errors on the NIPS semi-synthetic dataset as in Figure 2 (top, left). The $\ell_2$ norms of the topics were normalized to [0,1] for the computation of the $\ell_2$ error.

indicates that the LDA moments are somewhat sensitive to the choice of $c_0$. For example, the recovery $\ell_1$-error doubles when moving from the correct choice $c_0 = 1$ to the plausible alternative $c_0 = 0.1$ for $K = 10$ on the *LDAfix(200)* dataset (JD-LDA(10) line on the right of Figure 5).

### G.5  Comparison of the $\ell_1$- and $\ell_2$-errors

The sample complexity results [3] for the spectral algorithm for the LDA moments allow straightforward extension to the GP/DICA cumulants, if the results from Proposition 3.1 are taken into account. The analysis is, however, in terms of the $\ell_2$-norm. Therefore, in Figure 6, we provide experimental comparison of the $\ell_1$- and $\ell_2$-errors to verify that they are indeed behaving similarly.

### G.6  Evaluation of the real data experiments

For the evaluation of topic recovery in the real data case, we use an approximation of the log-likelihood for held out documents as the metric. The approximation is computed using a Chib-style method as described by [16] using the implementation by the authors.[18] Importantly, this evaluation methods is applicable for both the LDA model as well as the GP model. Indeed, as it follows from Section 2 and Appendix B.1, the GP model is equivalent to the LDA model when conditioning on the length of a document $L$ (with the same $c_k$ hyper parameters), while the LDA model does not make any assumption on the document length. For the test log-likelihood comparison, we thus treat the GP model as a LDA model (we do not include the likelihood of the document length).

### G.7  More on the real data experiments

The detailed experimental setup is as follows. Each dataset is separated into 5 training/evaluation pairs, where the documents for evaluation are chosen randomly and non-repetitively among the folds (600 documents are held out for KOS; 400 documents are held out for AP; 450 documents are held out for NIPS). Then, the model parameters are learned for a different number of topics. The evaluation of the held-out documents is performed with averaging over 5 folds. In Figure 3 and Figure 7, on the y-axis, the predictive log-likelihood in bits averaged per token is presented.

In addition to experiments with AP and KOS in Fig. 3, we demonstrate one more experiment with the NIPS dataset in Figure 7 (right).

Note that, as the LDA moments require at least 3 tokens in each document, 1 document from the NIPS dataset and 3 documents from the AP dataset, which did not fulfill this requirement, were removed.

Figure 7: Experiments with real data. **Left:** the KOS dataset. **Right:** the NIPS dataset. *Note*: a higher value of the log-likelihood is better.

Importantly, we observed that VI when initialized with the output of the JD-GP is consistently better in terms of predictive log-likelihood. Therefore, the new algorithm can be used for more clever initialization of other LDA/GP inference methods.

We also observe that the joint diagonalization algorithm for the LDA moments is worse than the spectral algorithm. This indicates that the diagonal structure (41) and (42) might not be present in the sample estimates (43) and (44) due to either model mis-specification or to finite sample complexity issues.

## Supplementary References

[31] A. Anandkumar, D.P. Foster, D. Hsu, S.M. Kakade, and Y.-K. Liu. A spectral algorithm for latent Dirichlet allocation. *Algorithmica*, 72(1):193–214, 2015.

[32] B.A. Frigyik, A. Kapila, and M.R. Gupta. Introduction to the Dirichlet distribution and related processes. Technical report, University of Washington, 2010.

[33] N. Halko, P.-G. Martinsson, and J.A. Tropp. Finding structure with randomness: probabilistic algorithms for constructing approximate matrix decompositions. *SIAM Rev.*, 53(2):217–288, 2011.

[34] T.G. Kolda and B.W. Bader. Tensor decompositions and applications. *SIAM Rev.*, 51(3):455–500, 2009.

[35] H.M. Wallach, D. Mimno, and A. McCallum. Rethinking LDA: why priors matter. In *NIPS*, 2009.