[Reviews · NeurIPS 2015]

Submitted by Assigned_Reviewer_1

I have read the author response to reviewer comments.

The authors cast LDA as discrete ICA and derive related spectral bounds and algorithms.

The discrete ICA view in eqn 5 and associated moment derivations on pg 2 are natural and straight forward, following directly on how spectral methods are typically derived for mixtures with a few Poisson-distributed related computations.

The diagnonalization algorithm also draws on existing approaches, though the authors note that the JD approach hasn't been applied in the LDA case.

The important point, I believe, is analysis that shows that this view has lower sample complexity and the related experiments to show this is the case.

Sample complexity is a major issue for moment-based approaches--limiting their application to moderately-sized data sets--and thus I feel like that part is extremely useful.

I'm also a huge fan of the very detailed derivations in the appendicies, which will be useful for many other applications.

Summary: Derivations are straight-forward but useful with a nice link to the empirical results.

Submitted by Assigned_Reviewer_2

This paper relates the problem of estimating Latent Dirichlet Allocation (LDA) to discrete ICA. The authors propose a joint diagonalization (JD) algorithm based on observed cumulants to learn the LDA parameters. Some experimental analysis comparing JD and other spectral methods such as tensor power iteration are provided.

The basic ideas for moment matching and forming higher order moments (cumulants) for learning ICA and LDA are not new, and already known. The authors only provide a form for the discrete version of ICA and relating to LDA. The JD algorithm is also not fundamentally novel.

The paper presentation can be also improved. The introduction is only a single paragraph which is very unusual.

It is useful to provide a figure for the LDA model, while only a series of different probabilistic models are provided in page 2 of the paper, which is hard for someone not familiar with these models to track them.

There are many typos: i.e. --> i.e., low of total ... --> law of total ... Note, that --> Note that \hat{D} --> \tilde{D}
Summary: The contribution of the paper is marginal. Many of these ideas for moment matching of latent variable models such as LDA are already known. The presentation of the paper is also not satisfactory.

Submitted by Assigned_Reviewer_3

The paper builds on the moment matching methods for standard LDA, providing related algorithms that estimate topic token distributions in a fully specified generative LDA model (one with a specific prior distribution for document length).

I found the paper quite clear, the approach interesting, and the results convincing. My only suggestion is to clean up the notation a bit - the zoo of models, variables, and indices makes it more difficult to parse than it should be.

Specifically:

I don't think the Discrete PCA section is necessary.

It seems like the GP model is just a stepping-stone to the more general DICA model for which results are presented, linking DICA to the natural choice of Poisson distribution for L.

However, it might be more clear to skip directly to the DICA model, and bring up earlier the meaning of alpha's (topic intensities).

At the end of section 4, the authors mention that the violation of the non-negativity constraint is practically significant.

What consequence does that have on convergence, stability?

This issue seems important enough to expand a bit.

Regarding the last point made in Conclusions, is it at least possible to estimate the relative scales of topic intensities given assumptions on D?

Minor things:

line 137, 669 "low of total" to "law of total"

line 298 "some up" to "sum up"

line 366 we set b in (4)(?)
Summary: This is a fairly clear paper that brings together results from several threads, builds on previous analytical derivations to provide methods for estimating parameters in a more complete generative LDA model, and demonstrates improved data efficiency and robustness on real and synthetic data.

Author Feedback
Author rebuttal: We thank the reviewers for their time and their valuable feedback. For the final version, we will improve the presentation as suggested. We address below some of their additional concerns.

R2: [contributions]
Indeed, the method of moments is not new and has already been applied to ICA and LDA (see refs in the paper). However, given the popularity of LDA in machine learning, we believe that our combination of several threads in this setting makes valuable contributions. More specifically:

- We highlight the close connection between DICA and LDA, enabling the re-use of important ideas from the ICA literature.
- We derive the cumulants for DICA, which are not the same as cumulants for ICA. Using cumulants instead of non-central moments makes derivations more straightforward and formulas more compact.
- We show theoretically and empirically that the (DICA/GP) cumulants have better sample complexity than the standard (LDA) moments.
- We apply the JD algorithm to the problem, for the first time to the best of our knowledge, and demonstrate its state-of-the-art performance.
- We perform an extensive comparison of different method of moments algorithms for LDA/DICA, which we believe is important for further research in the field.
- Note that the new algorithm is computationally very efficient. The overall cost of the whole algorithm is roughly the same as one iteration of variational inference.

Therefore, although the new algorithm is partially composed of known parts, we believe that making these connections is a useful contribution to the topic modeling literature and leads to a more robust algorithm, both theoretically and empirically.

R6: [log-likelihood]
In Section 2 (and Appendix A), we explain that the GP model is equivalent to the LDA model when conditioning on the length of a document (with the same c_k hyperparameters). For the test log-likehood comparison, we thus treat the GP model as a LDA model (we do not include the likelihood of the length of the document).

R4/R6: [real data experiments]
In the submitted paper we had two experiments on real data (AP and KOS in the appendix) and made another experiment with the NIPS data set in the meantime. We observe that sometimes our new algorithm (JD-GP) gives better log-likelihood than variational inference (VI) and sometimes slightly worse. There could be different reasons for that:
- VI is (a) an approximation and (b) solves a non-convex problem, i.e. does not find the global optimum.
- The topics estimated by GP are the same as DICA, which makes less assumptions about the prior distribution on the topic proportions than LDA (only independence), and so could potentially work better than LDA, e.g., when the distribution is not unimodal.
- We found that the predictive log-likelihood for the moment matching algorithms are somewhat sensitive to the way the hyperparameters of the topic mixture distribution are estimated (see also R3 [topic intensities] below).

Importantly, we observed that VI when initialized with the output of the JD-GP is consistently better in terms of preditive log-likelihood and (at least few times) faster than VI with a random initialization. Therefore, the new algorithm can be used for more clever initialization of other LDA/GP inference methods.

R3/R4: [non-negativity of topic matrix]
In theory, if we have infinite amount of data from the correct model (LDA/GP), no negative values appear in the estimated topic matrix and we observed this experimentally. This is not the case in the presence of noise (either through sampling or through model misspecification). We empirically found strong correlation between a measure of negativity of the matrix, the recovery (L1) error, and the amount of noise (increasing noise increases the others), which is why we mention that the negativity issue is potentially important. As mentioned in lines 248-250, we investigated a joint diagonalization algorithm that also penalized negative values for the pseudo-inverse of the diagonalizing matrix. We only achieved negligible improvements in terms of recovery error, with a much higher computational cost, and thus decided not to cover it in the paper due to the space restriction.

R3: [stability]
Importantly, even in the presence of heavy noise, we did not observe any convergence stability issues for the JD algorithm (as opposed to others, such as TPM -- see right plot of Figure 1). We are not aware of theoretical guarantees for its stability however.

[topic intensities]
We can indeed estimate the hyperparameters of the prior over topic intensities from its cumulants appearing as the diagonal terms in equation (12) and (14). In fact, we did this for the predictive log-likelihood experiments in Figure 3 (we will add the precise description to the final version). Other standard approaches, such as variational inference, ML, or MAP, can also be used. Detailed comparison of these is a question for future research.